# Wilson Disease: Copper-Mediated Cuproptosis, Iron-Related Ferroptosis, and Clinical Highlights, with Comprehensive and Critical Analysis Update

**DOI:** 10.3390/ijms25094753

**Published:** 2024-04-26

**Authors:** Rolf Teschke, Axel Eickhoff

**Affiliations:** 1Department of Internal Medicine II, Division of Gastroenterology and Hepatology, Klinikum Hanau, D-63450 Hanau, Germany; eickhoff.axel@gmx.de; 2Academic Teaching Hospital of the Medical Faculty, Goethe University Frankfurt, D-60590 Frankfurt, Germany

**Keywords:** Wilson disease, copper overload, cuproptosis, ferroptosis, genetic testing, family screening, Haber–Weiss reaction, Fenton reaction, reactive oxygen species (ROS)

## Abstract

Wilson disease is a genetic disorder of the liver characterized by excess accumulation of copper, which is found ubiquitously on earth and normally enters the human body in small amounts via the food chain. Many interesting disease details were published on the mechanistic steps, such as the generation of reactive oxygen species (ROS) and cuproptosis causing a copper dependent cell death. In the liver of patients with Wilson disease, also, increased iron deposits were found that may lead to iron-related ferroptosis responsible for phospholipid peroxidation within membranes of subcellular organelles. All topics are covered in this review article, in addition to the diagnostic and therapeutic issues of Wilson disease. Excess Cu^2+^ primarily leads to the generation of reactive oxygen species (ROS), as evidenced by early experimental studies exemplified with the detection of hydroxyl radical formation using the electron spin resonance (ESR) spin-trapping method. The generation of ROS products follows the principles of the Haber–Weiss reaction and the subsequent Fenton reaction leading to copper-related cuproptosis, and is thereby closely connected with ROS. Copper accumulation in the liver is due to impaired biliary excretion of copper caused by the inheritable malfunctioning or missing ATP7B protein. As a result, disturbed cellular homeostasis of copper prevails within the liver. Released from the liver cells due to limited storage capacity, the toxic copper enters the circulation and arrives at other organs, causing local accumulation and cell injury. This explains why copper injures not only the liver, but also the brain, kidneys, eyes, heart, muscles, and bones, explaining the multifaceted clinical features of Wilson disease. Among these are depression, psychosis, dysarthria, ataxia, writing problems, dysphagia, renal tubular dysfunction, Kayser–Fleischer corneal rings, cardiomyopathy, cardiac arrhythmias, rhabdomyolysis, osteoporosis, osteomalacia, arthritis, and arthralgia. In addition, Coombs-negative hemolytic anemia is a key feature of Wilson disease with undetectable serum haptoglobin. The modified Leipzig Scoring System helps diagnose Wilson disease. Patients with Wilson disease are well-treated first-line with copper chelators like D-penicillamine that facilitate the removal of circulating copper bound to albumin and increase in urinary copper excretion. Early chelation therapy improves prognosis. Liver transplantation is an option viewed as ultima ratio in end-stage liver disease with untreatable complications or acute liver failure. Liver transplantation finally may thus be a life-saving approach and curative treatment of the disease by replacing the hepatic gene mutation. In conclusion, Wilson disease is a multifaceted genetic disease representing a molecular and clinical challenge.

## 1. Introduction

All heavy metals, from aurum (gold) to zinc including copper (Cu from Latin cuprum), are found on earth with various amounts, but they all were originally formed in the universe from helium and hydrogen via nuclear fusion inside stars during supernova explosions millions of years ago [1,2,3,4]. On our globe, copper entered the flora and became an essential micronutrient for plants, animals, and humans, who may be confronted with metabolic disorders that arise from mishandling of copper [5]. In plants, copper is a cofactor for a variety of enzymes, and it plays an important role in signal transduction, the antioxidant system, respiration, and photosynthesis [5,6]. Several mechanisms ensure copper homeostasis in plants [6], modify the diurnal and circadian clock rhythms [7], and take care of abiotic plant stress due to copper [8]. Copper found in plants [5,6,7] is taken up by the plants from the soil [8], and these copper containing plants are the first in the food chain providing cattle grazing on the fields with plant-derived copper before humans consume the plants and meat.

Copper contaminating food and drinking water can easily enter the human body, but generally does not harm healthy individuals as long as they are not compromised by *ATP7B* gene mutations that disturb copper homeostasis, like in Wilson disease. This mutation primarily impairs the biliary excretion of excess copper that initiates cuproptosis. Due to the concomitant deposition of iron in the liver, ferroptosis may contribute to liver injury. The consensus exists that both cuproptosis and ferroptosis cause special forms of regulated or programmed cell death. Because Wilson disease is a rare hereditary disorder with interesting molecular and mechanistic aspects, further analyses and discussions are warranted.

The current article critically analyses developments of the pathophysiology including cuproptosis, ferroptosis, and ROS in patients with Wilson disease. It also discusses its gene mutations, clinical features, symptoms, diagnostic criteria, drug therapy with copper chelators, prognosis, and possible liver transplantation as ultima ratio.

## 2. Copper Homeostasis in Healthy Humans

### 2.1. Quantitative Aspects of Copper

In human biological samples, copper is available as soluble Cu^2+^ ion, termed as cupric, and as Cu^1+^ ion, known as cuprous that is insoluble [9,10]. To overcome the insolubility, cuprous is commonly found as complex with other biomolecules [9,11]. A balanced diet commonly provides copper in amounts of around 1.2–1.3 mg/day [9], or slightly higher with amounts of around 1.5 mg/day [10]. Around 0.8 mg are daily absorbed from the stomach and the small intestine [9,10,12], sufficient to maintain in a healthy human a total body store of around 80 mg copper with 10% thereof confined to the liver [10]. The current adult recommended dietary allowance (RDA) for copper is 1 mg/day [13], and the requirement for extra copper during pregnancy is with 100 µg per day relatively low [14].

### 2.2. Physiological Role of Copper in Humans

Copper is an indispensable trace metal element in humans due to its multifaceted involvement in mitochondrial energy production and maintenance, redox homeostasis, synthesis of bio-compounds, as a general signaling molecule, and in extracellular matrix modeling [10,15,16,17]. In particular, copper in physiological amounts is incorporated in hemoglobin formation and plays a role in drug and xenobiotic metabolism, carbohydrate metabolism, catecholamine biosynthesis, cross-linking of collagen, elastin, hair keratin, and antioxidant defense mechanisms [10]. Conducting these functions, various copper-depending enzymes are involved, such as oxygenases and oxidoreductases, allowing for electron transfer [10,15]. Among these, also, are hydroxylases, transferases like cytochrome c oxidase, Cu/Zn superoxide dismutase, hephaestin, ceruloplasmin, ferroxidases, monoamine oxidase, lysyl oxidase, and dopamine β-monooxygenase catalyzing the conversion of dopamine to epinephrine [10,15,16,17].

Copper deficiency and sideroblastic anemia are causes of several diseases that include neurodegenerative diseases, cancer, cardiovascular disease [16], and Menkes disease [16,18,19]. As opposed, excess copper accumulation is characteristic of Wilson disease [16,18], and also with iron deposits detectable in the liver of some patients. Clinical copper deficiency due to consumption of low-copper food or impaired gastrointestinal copper absorption, and is well known, with clinical signs mainly being hematological and neurological manifestations including leukopenia, microcytic, normocytic, or macrocytic anemia, myelopathy, and peripheral neuropathy [20,21]. It also was described as a triad consisting of anemia, leucopenia, and myeloneuropathy [22]. Copper maldistribution in some tissues is associated with low copper levels in the heart, liver, muscle, skin, and connective tissues that are causally related to mutations in the *ATP7A* gene of Menkes disease, whereby ATP7A represents an ATPase that transports copper [18,19]. The resulting loss of functional ATP7A in the intestine of patients with Menkes disease leads to an accumulation of copper in enterocytes and follows reduced copper efflux into the blood [23]. Menkes disease commonly has a fatal clinical course with deaths in early childhood [16,17,18,19,23], supported by studies in animals with genetic defects in transport of copper to critical sites for development and function, and is associated with early death [24].

### 2.3. Gastrointestinal Absorption of Copper from Food

Copper contained in food and water enters the human body via the epithelial cells of the gastrointestinal tract through the copper transporter 1 (CTR1) protein [16,25] that also is important for copper homeostasis in the liver [25,26,27,28,29,30]. The enterocytic process is facilitated by the activity of the metalloreductases six-transmembrane epithelial antigen of the prostate (STEAP) and duodenal cytochrome b (DCYTB), which reduce divalent Cu^2+^ to monovalent Cu^1+^, the ionic state in which CTR1 transports copper [16]. The CTR1 protein helps move copper inside the enterocytes where copper is bound to metallothionein or carried by the antioxidant protein 1 (ATOX1) [16,25,26,27,28,29,30]. After incorporation into the enterocytes, copper ions are preferentially delivered to basolateral ATP7B, which transports Cu^1+^ to the apical membrane and pumps it from the enterocytes into the blood.

### 2.4. Transfer of Copper from Enterocytes into the Circulation

Enterocytes become preferentially liberated from high amounts of copper through their cellular basolateral transmembrane ATP7B and slightly by ATP7A, which facilitates the release of copper into the portal vein as bound to albumin before entering the liver [25,26,31]. *ATP7A* and *ATP7B* gene expression patterns are partially complementary throughout the human body. The *ATP7A* gene is expressed in the majority of tissues except for the liver. *ATP7B* gene expression is more restricted, with highest expression in the liver, and increased co-expression with the *ATP7A* gene in many cell types and tissues is becoming evident [32]. The intracellular localization and copper-regulated trafficking of ATP7A and ATP7B reflects their biosynthetic and protective roles in cellular copper homeostasis [31,32]. Secreted in the bloodstream, copper is bound to soluble macroglobulins [16].

### 2.5. Copper in the Liver of Healthy Humans and Genetic Control of Internal Copper Environment

After arriving at the liver, copper is taken up by the hepatocytes via the membrane copper transporter CTR1 [16]. Inside the cytoplasm, copper is then either delivered to specific proteins or chelated by metallothionein (MT) for storage [16]. The principal copper chaperones include the cytochrome c oxidase copper chaperone (COX1) that delivers copper to cytochrome c oxygenase, copper chaperone for superoxide dismutase (CCS), which provides copper to superoxide dismutase 1, and ATOX1, which supplies copper to ATP7B [16]. In the liver, ATOX1 binds copper inside the hepatocytes. Once here, ATP7B links the copper to an organelle system known as the trans-Golgi network (TGN) [25,26,27,28,29,30], where both ATPases, ATP7A and ATP7B, normally reside, but with increasing intracellular copper levels, they move to cytoplasmic vesicles nearby the cell periphery, where their primary role is in the excretion of the excess copper from the epithelial cells of the gastrointestinal tract [31].

The ATPase ATP7B in the hepatocytes helps ceruloplasmin acquire copper in the trans-Golgi network (TGN) and facilitates the delivery of copper to the apical bile canaliculus for export of copper via the biliary tract or pump copper ions from the liver back into the blood, where it again binds to soluble chaperones and is transported to specific tissues and organs [32,33]. Upon reaching its target tissues, copper catalyzes reactions in a wide range of physiological processes, including mitochondrial energy production, tyrosine and neurotransmitter metabolism, redox homeostasis, and extracellular matrix remodeling [32,33,34,35].

Copper homeostasis in healthy individuals is under genetic control, ensuring appropriate copper amounts in all organs, and thereby contributing to human health by avoiding excess of total copper body stores and preventing low stores [9,14,15,16,17] involving the *ATP7B* gene [18,19] described as the copper-transporting gene that encodes the transmembrane copper-transporting P-type ATPase ATP7B [19]. The *ATP7B* gene has been mapped to chromosome 13q14.3 [20], and contains 20 introns and 21 exons for a total genomic length of 80 kb [20,21,22]. Itis synthesized in the endoplasmic reticulum of the hepatocytes, and is then relocated to the TGN within hepatocytes [19], where ATP7B functions in either the TGN or in cytoplasmic vesicles. In the context of TGN, ATP7B activates ceruloplasmin by packaging six copper molecules into apoceruloplasmin, which then is secreted into the plasma. In the cytoplasm, ATP7B sequesters excess copper into vesicles and excretes it via exocytosis across the apical canalicular membrane into bile [19,20,22,23,24,25,36,37,38,39,40].

## 3. Basics of Wilson Disease

### 3.1. ATP7B Gene Mutation

Wilson disease is an autosomal recessive disorder based on variants of the *ATP7B* genes that encode the protein ATP7B [26]. The majority of disease-associated mutations in *ATP7B* genes are missense mutations, and consensus exists that *ATP7B* genes are located on 13q14.3. Due to the dualistic role of ATP7B in both the synthesis of the copper-containing protein ceruloplasmin and biliary excretion of copper, defects in its function lead to copper accumulation in the liver and other organs that express ATP7B [16,32,40] because both functions of ATP7B are impaired in Wilson disease due to *ATP7B* gene mutation [25,26,31,33,41]. As a consequence, apoceruloplasmin is secreted in a form that lacks copper, and its circulating half-life is shorter than the holoprotein with a proper complement of copper, leading to lower concentration of the protein in the circulation [25]. ATP7B is most highly expressed in the liver, but is also normally found in the kidney, placenta, mammary glands, brain, and lungs [42]. These organs contain either Cu^2+^ or Cu^1+^. Excess amounts of copper in these and other organs explains the broad manifestation of Wilson disease due to local *ATP7B* gene mutations [43].

### 3.2. Pathophysiology of the Copper Liver Injury

The excessive hepatic copper accumulation, which is substantially well-above physiological amounts found in healthy individuals, leads in patients with Wilson disease to develop significant liver injury [44]. A broad range of proposals regarding how copper mechanistically causes the liver injury have been published [45,46,47,48,49,50]. Mechanistic events are similar in animals or humans exposed to excessive amounts of exogenous copper [50,51] compared to patients with Wilson disease, except for their *ATP7B* gene mutations as a basis for this specific clinical liver injury [25,45]. The cascade of molecular, mechanistic, and pathogenetic steps leading to the final liver injury of Wilson disease is listed in condensed form to provide a quick overview (Table 1) [10,15,16,25,26,31,33,40,41,42,52,53,54,55,56,57,58,59,60,61,62,63,64,65,66,67,68,69,70].

There are several steps ultimately leading to the liver injury in Wilson disease as outlined in short above (Table 1). The mechanistic steps are extremely complex requiring a careful discussion and analysis in detail.

#### 3.2.1. Intestinal Copper Uptake

The first step is confined to the intestinal uptake of copper in normal amounts originating from the food chain that contains normal copper quantities [16,25]. The extent of normal copper absorption is genetically controlled and regulated by demand to ensure copper homeostasis in the organs. Copper absorption proceeds via specific copper transporters located in the enterocytes [71].

#### 3.2.2. Copper Transfer from Enterocytes to Hepatocytes

The second step involves the subsequent hepatocellular uptake of copper provided to the respective copper uptake transporters of the liver cell membrane. For this purpose, copper is delivered from the copper binding plasma proteins such as albumin, ceruloplasmin, and transcuprein/alpha-2-macroglobulin [71].

#### 3.2.3. Copper Overload Due to Impaired Biliary Copper Excretion

The third step represents the genetic impairment of biliary copper excretion [25]. This ATP7B functional impairment causes copper overload primarily in the liver and other organ accumulation afterwards [43]. The broad deposition of copper explains the multifaceted clinical characteristics of Wilson disease.

#### 3.2.4. Role of ROS and Vicious Cycles

The fourth step of cascade events in Wilson disease contributing to the liver injury is the most challenging one, because the liver of some patients contains not only excessive amounts of copper [56,72], but also considerable quantities of iron [54,56,60,72], with iron also found in the brain providing conditions linked to ferroptosis in Wilson disease [73]. Iron accumulation is not generally a major feature of Wilson disease just due to a low ceruloplasmin level, because most patients do have some circulating level of oxidase-active ceruloplasmin. If present, iron deposition is linked to the known decreased level of ceruloplasmin [72] that exerts ferroxidase activity causing tissue storage of iron, like in patients with hemochromatosis [56]. Both metals have the production of ROS in common that is associated with injurious properties [56,60]. These processes are known as cuproptosis for copper [56,57,61,62] and ferroptosis for iron [56,62]. The complexity of cuproptosis and ferroptosis is also highlighted by the generally accepted view that the ROS primarily generated by copper or iron ions may proceed via the Haber–Weiss reaction and/or Fenton reaction [74,75,76]. Due to these current uncertainties, the various possible reaction types are provisionally listed (Table 2) [74,75,76,77,78].

Copper plus hydrogen peroxide produces, with hydroxyl radicals, the main ROS (Table 2), as earlier evidenced by electron paramagnetic resonance (EPR) spectroscopy measurements [78]. This is facilitated by the combination of the Haber–Weiss reaction and the Fenton reaction providing, as a net result, ROS with radical types identical to those created by iron (Table 2) [74,75,76,77,78]. Even worse is that both metals now may each create a vicious cycle: Cu^2+^ is reduced to Cu^1+^, which in turn is converted back to Cu^2+^, whereas Fe^3+^ is oxidized to Fe^2+^, which in turn is converted back to Fe^3+^. It is obvious that both vicious cycles likely contribute to the liver injury perpetuation through an ongoing, ultimately irreversible chain reaction mechanism, with a higher risk elicited by the higher hepatic levels of copper as compared with iron.

#### 3.2.5. Cuproptosis and Ferroptosis

The fifth step leading to the liver injury combines both, cuproptosis related to copper [58,62,79] and ferroptosis related to iron [59,60,80,81,82,83,84,85,86,87]. The consensus exists that cupro-ptosis and ferroptosis cause special forms of regulated cell death (RCD), also known as programmed cell death (PCD) [62,79], that represent regulable processes governed by well-organized signaling cascades and molecular mechanisms [62]. Both forms are closely associated with each other and have common pathways including oxidative stress, but exhibit distinct morphological and biochemical features compared with other cell-death forms such as apoptosis, autophagy, pyroptosis, and necroptosis [62]. Oxidative stress and/or proteasome inhibition promote apoptosis, autophagy, and pyroptosis, while lipid peroxidation is involved in necroptosis.

##### Cuproptosis

Cuproptosis as the newly described copper-dependent form of RCD [58] that is considered as contributory causative mechanism in Wilson disease [79], characterized by protein aggregation, protein ubiquitination, and DNA damage [62]. At the molecular level, this RCD form is mechanistically triggered by the Haber–Weiss and Fenton reactions (Table 2). In more detail, excess intracellular copper in hepatocytes, like in Wilson disease, induces the aggregation of lipoylated dihydrolipoamide S-acetyltransferase (DLAT) [58]. This is associated with impairment of the mitochondrial tricarboxylic acid (TCA) cycle, resulting in proteotoxic stress in line with the proposed cuproptosis [58,62]. In human cells, the copper-dependent RCD is distinct from other known death mechanisms and is dependent on mitochondrial respiration [62]. Copper induces not only DLAT aggregation [58], but also binds to DLAT, from which the cell dies due to proteotoxic stress [62]. It is also noteworthy that lipoylated protein aggregation is associated with subsequent iron–sulfur cluster protein loss [62], possibly releasing iron and adding to the existing iron accumulation in the liver of the patient with Wilson disease. This would support the close relationship between copper and iron in liver injury [79].

##### Ferroptosis

Ferroptosis as the iron-dependent form of RCD that is induced by iron accumulation and excessive phospholipid peroxidation [62], also classified as non-apoptotic cell death [80]. At the molecular level and similar to cuproptosis, ferroptosis is initiated by the Haber–Weiss and Fenton reactions in a mechanistic cascade of events (Table 2). Accumulated iron, like in hemochromatosis, injures membranes of subcellular organelles such as mitochondria or the endoplasmic reticulum [59,81]. They contain structural phospholipids rich in polyunsaturated fatty acids (PUFAs) that are peroxidized [81]. This is supported by lipid peroxide markers found in patients with hemochromatosis or overloaded by exogenous iron [81,82,83,84,85]. As an example, liver biopsy specimens obtained from patients with hemochromatosis were immunostained for protein adducts with malondialdehyde (MDA) and 4-hydroxynonenal [86]. Both adducts were more abundantly found compared to controls, whereby the staining had a predominance in acinar zone 1 that followed the localization of iron [86]. Similarly, enhanced oxidative stress was described in patients with hemochromatosis, as evidenced by hepatic MDA protein adducts and increased oxidatively modified serum proteins [86]. MDA-lysine epitopes and oxidatively modified serum proteins, as well as immunoglobulin G autoantibodies against MDA-lysine epitopes, were increased in untreated hemochromatosis patients, compared to normal individuals [83]. Whether iron-related MDA processes are involved in Wilson disease has not yet thoroughly been studied in the liver of patients with this disease in relation to iron deposits. However, increased MDA levels were reported in the blood of patients with Wilson disease, lacking a description of whether these MDA results were obtained from patients with iron-loaded livers [87]. In any event, and as newly conceptualized, iron and copper metabolisms cross over different metabolic pathways leading to a disrupted iron metabolism in patients with Wilson disease that may affect the clinical course of the disease [60].

#### 3.2.6. Cross-Talk of Inflammatory Cytokines

As the sixth step in the further course of Wilson disease, it seems that hepatic immune cells become more involved, as shown by the detection of inflammatory mediators like cytokines in the plasma of patients with Wilson disease [63,87,88]. This is an interesting observation, because it converts a silent liver to an organ that helps transfer of information from the injured human liver with its mechanistic processes to the blood of patients, easily available for further analysis by physicians and scientists [63]. The study cohorts were composed of 99 patients with Wilson disease and 32 healthy controls. Compared with controls, in patients with Wilson disease, there was a significant increase of plasma of T helper (Th) 1 cells (IL-2, TNF-α, and TNF-β), Th2 cells (IL-5, IL-10, and IL-13), and Th17 (IL-23) (*p* < 0.05). Higher plasma Th 1 cells (IL-2, TNF-α, and TNF-β), Th 2 cells (IL-13), and Th 17 (TGF-β1, IL-23) levels were found in neurological patients compared with control groups (*p* < 0.01). In addition, Th 1 cells (TNF-α and TNF-β), Th 3 (TGF-β1), and Th 17 (IL-23) levels were significantly higher in hepatic and neurological patients (*p* < 0.05), whereby the higher Th1 cells (IL-2, TNF-α, and TNF-β), Th2 cells (IL-13), Th17 (TGF-β1, IL-23), and the course of Wilson disease were associated with the severity of the neurological symptoms for patients with Wilson disease [63]. Similar results of inflammatory mediators, such as cytokines have been reported in other cases of Wilson disease [87,88]. The multiplicity of blood mediators reflects substantial cross-talking among each other in the liver of patients with Wilson disease, but it appears premature to define the action or function of each mediators because they may stimulate or inhibit functions of the others and confound the results [63,87,88].

These cautious considerations are in line with the proposed interaction of copper with immune-cell-producing mediators [67,68,69], explicitly shown by the reduced interleukin-2 (IL-2) production and IL-2 mRNA in human T-lymphocytes caused by copper deficiency [69]. It is also known that prolonged exposure of the liver to high amounts of copper will activate resident hepatic stellate cells to myofibroblasts, which secrete extracellular matrix proteins that generate collagen [64]. This may lead to liver fibrosis and cirrhosis [65,89].

Despite some uncertainties, it remains to be established whether animal models [90,91], including the goldfish model [92] or the zebrafish model [93], can contribute to closing the existing mechanistic gaps of excess copper liver injury.

#### 3.2.7. Gut Microbiome and Dysbiosis

Fifth, recent evidence suggests a correlation between dysbiosis in gut microbiome and multiple diseases [70,94,95,96,97] such as alcoholic liver disease [95] and genetic heavy metal liver disorders including hemochromatosis [96,97] and Wilson disease [70]. As an example, 16S rRNA sequencing was performed on fecal samples from 14 patients with Wilson disease and was compared to the results from 16 healthy individuals [70]. The diversity and composition of the gut microbiome in the Wilson disease group were significantly lower than those in healthy individuals. The Wilson disease group presented unique richness of *Gemellaceae*, *Pseudomonadaceae*, and *Spirochaetaceae* at family level, which were hardly detected in healthy controls. This group had a markedly lower abundance of *Actinobacteria*, *Firmicutes*, and *Verrucomicrobia*, and a higher abundance of *Bacteroidetes*, *Proteobacteria*, *Cyanobacteria*, and *Fusobacteria* than that in healthy individuals. The *Firmicutes* to *Bacteroidetes* ratio in the Wilson disease group was significantly lower than that of healthy controls. In addition, the functional profile of the gut microbiome from Wilson disease patients showed a lower abundance of bacterial groups involved in the host immune- and metabolism-associated system pathways, such as transcription factors and ABC-type transporters, compared to healthy individuals. These results implied that the dysbiosis of gut microbiota may be influenced by the host metabolic disorders of Wilson disease, which provides a new understanding of the pathogenesis, and even new possible therapeutic targets for patients with Wilson disease. However, the impact of intestinal microbiota polymorphism in Wilson disease have not yet fully been elaborated, and need to be explored for seeking some microbiota benefit for Wilson disease patients [70]. Gut dysbiosis is directly related with increased intestinal permeability as a consequence of epithelial barrier deterioration, tight junction alteration, and bacterial translocation, and these factors cause endotoxemia, which might reach and damage the liver through the portal vein [70,94,95]. A flow chart of key molecular pathways leading to Wilson disease is presented (Figure 1).

### 3.3. Natural Clinical Course of Wilson Disease

As a copper-storing liver disease caused by inheritable malfunctioning or missing *ATP7B* genes, Wilson disease is characterized by disturbed cellular homeostasis of copper handling primarily in the liver cell [89]. If copper export from the liver increases after storage capacity is exceeded, the toxic copper enters the circulation and arrives at other organs, causing local accumulation and cell injury. This explains why copper injures not only the liver, but is also responsible for liver unrelated symptoms and various clinical features. The natural clinical course could include neurologic and/or psychiatric disease. In the absence of a specific treatment, the natural course of Wilson disease is deleterious, and may end up with acute liver failure, cirrhosis, and, rarely, hepatocellular carcinoma [66,89].

### 3.4. Differential Diagnosis of Wilson Disease

Apart from Wilson disease, many conditions exists that are related to exposure with high amounts of exogenous copper, requiring thorough exclusion of differential diagnoses [51,98,99,100,101,102]. They require special clinical attention regarding diagnostic and therapeutic approaches. Examples are acute copper intoxications and chronic copper poisonings, including the worldwide highly disputed erroneously diagnosed and described Indian childhood cirrhosis (ICC). Careful clinical and toxicological studies finally led to a different view of the disease complex, subsequently termed Indian childhood cirrhosis-like disorder or better idiopathic childhood cirrhosis syn idiopathic copper toxicosis, with clinical and pathophysiological features substantially different from those of the classic Wilson disease. Differential diagnosis also should include other acute and chronic liver diseases [47], glycosylation defects with cholestasis, and progressive familiar intrahepatic cholestasis (PFIC) variants, and should not forget the neurological and psychiatric presentations of the disease.

#### 3.4.1. Acute Copper Poisoning

Acute copper poisoning is more often observed in South Asian countries where it is most prevalent in rural populations [51], while it is uncommon in Western countries [98]. Copper sulfate is easily available worldwide, and is used in agriculture, leather industry, and at home to make glue [51]. In addition, the burning of copper sulfate at home or in shops, seen as a good luck charm and for religious reasons, is a widespread practice in countries like India where Buddhists and Hindus live [51,99]. Copper intoxications have been reported worldwide following accidental or intentional exposure through various routes, including orally [51,98,100,101,102]. Toxic concentrations occur after the ingestion of as little as 1 g [98], while the lethal dose of ingested copper is 10 to 20 g [51]. The toxic ingested dose is absorbed by the gastrointestinal tract, and is unlikely to be attached to ceruloplasmin [103]. What is absorbed will bind to albumin, and much will go through the liver into the bile, and excess will be released in urine where copper usually is only minimally excreted, but in this setting urine copper output can significantly be elevated.

Clinical manifestation of the acute copper intoxication shows, in a patient, substantial liver injury with serum aspartate aminotransaminase (AST) 2340 U/L and alanine aminotransaminase (ALT) 780 U/L, ratio AST/ALT of 3, and lacking serum alkaline phosphatase (ALP) results in the determination of the ratio R value to determine the liver injury pattern [98]. In another patient, serum ALT activity was 10 U/L with ALP activity of 406 U/L [99], in line with a cholestatic liver injury as based on laboratory data [104]. Acute liver failure can develop due to direct copper toxicity [99,100,101].

Evidence-based recommendations for treatment of acute oral intoxications by copper are not available, likely due to the small number of affected patients, which does not allow for valid randomized controlled trials (RCTs) evaluating the efficacy and risks of therapeutic approaches. As a result, therapeutic measures are currently based on experience, as described in anecdotal reports. Gastric lavage was viewed as usually unnecessary due to persistent vomiting [98]. Particular care is certainly needed for spontaneous vomiting due to the aspiration risk, especially in unconscious patients, who may require a preventive endotracheal tube (ETT) in analogy to acute oral intoxications by aliphatic halogenated hydrocarbons like carbon tetrachloride or chloroform [105,106,107]. If imaging data of abdominal ultrasound, computer tomography, or magnetic resonance data reveal a bezoar composed of dense particles of conglomerated ingested copper in the stomach, endoscopic removal should be considered in the patient, who needs ETT to prevent accidental respiratory aspiration of the metal product.

There is no place for intentional forced vomiting, and the use of activated charcoal was anecdotally reported despite the lack of efficacy [98]. Occasionally drinking milk and physiological saline was also recommended [99], but intestinal lavage may be a better option to quickly remove the copper out of the intestine, a procedure successfully applied in patients intoxicated by ingestion of aliphatic halogenated hydrocarbons [105,106,107]. Chelating agents like D-penicillamine are commonly applied and recommended in severe poisoning, although pharmacokinetic data are scarce to guide their use [98]. Occasionally zinc was also used [50]. Hemolysis-induced anemia may be corrected by transfusion of packed red blood cells, and hemodialysis may be indicated for renal insufficiency [99,108]. Renal failure will require hemodialysis to temporarily substitute the injured non-functional kidneys without the intention to remove the toxic copper from the blood [99]. Of note, reports claim that through contaminated hemodialysis fluid infusions, the copper may be freed from the dialysis device and given into the dialyzed patients [51].

Prognosis is poor unless patients receive quick treatment with chelation and supportive measures in face of lethality rates ranging from 14% to 36% [51]. There is no supporting evidence that acute copper ingestion increases the risk of hepatocellular carcinoma. Most importantly, stopping the over-the-counter sale of copper sulphate and restricting sales to authorized agents is strongly recommended to reduce the risk of acute copper intoxication, as is, alternatively, providing copper sulphate not as fine powder to be easily solved, but as large crystals [51,99]. Reports which describe acute liver injury in connection with exposure to environmental copper are lacking [46].

#### 3.4.2. Prolonged Copper Intoxication

In a patient with prolonged exogenous copper exposure for 10 years, serum activity of ALT was 51 U/L associated with AST 190 U/L and ALP 68 U/L, but lacking the upper limit of normal (ULN) data prevented the calculation of the R value [50]. The patient used up to an 8 mg copper daily dose for treating copper deficiency, known as human Swayback disease, whereby this iatrogenic copper overload led to liver transplantation due to compensated cirrhosis. No liver tests (LTs) values have been reported in patients with acute renal failure following copper sulphate intoxication [109]. Similarly, no ALT or AST values are available in copper workers [46,47].

In the patient with overdosed copper exposure for a long time, transjugular liver biopsy demonstrated, upon light microscopy, ongoing portal and segmental inflammation with ballooning degeneration of the hepatocytes, as well as diffuse hepatocellular copper accumulation on the rhodamine stain following prolonged copper overdose [50]. The diagnosis of cirrhosis was first established at an occasion of a laparotomy for umbilical hernia repair, as evidenced by macroscopical morphology, and later confirmed at the time of liver transplantation, showing both micronodules and macronodules of the liver surface. Analysis of hepatic content of copper provided extremely high values, suggesting copper as the causative agent in this patient.

Experimental studies in animals showed mostly unchanged, or in rare cases slightly increased, serum ALT and AST activities after application of high copper amounts [110,111,112]. In animals, however, overdosed copper administration caused no liver injury, as assessed by light microscopy [92,110,111], or only minimal, partially dose-dependent changes such as small vacuoles of hepatocytes, hepatocyte swelling, inflammatory cells, or sinusoidal congestion [112]. Electron microscopy data on the liver of the patient exposed to extreme amounts of copper were not available [50,109], but have been reported in animal studies as irregularly shaped nuclei, abundant mitochondria, and displayed cristae, and hepatocytes with inclusion of secondary lysosomes [110]. Prolonged intoxication with exogenous copper has to be differentiated clinically from Wilson disease [66,89,113], a genetic disorder of the liver leading to hepatic copper accumulation [114,115].

##### Indian Childhood Cirrhosis

A special challenge is the earlier termed ICC, which was attributed in previous reports preferentially to exogenous copper in drinking water or milk [116], but this obviously was an assumption error [117,118,119,120,121,122]. In more detail, ICC was initially considered preventable and to be clearly distinguishable in Indian children from other chronic liver disorders including Wilson disease [116]. Grossly increased hepatic, urinary, and serum copper concentrations were described as characteristic findings of ICC. These increased concentrations were easily demonstrated histologically with orcein-rhodamine staining. Environmental ingestion of copper appeared to be the most plausible explanation for ICC, as shown by feeding histories, the prevention of ICC in siblings and in the Pune district by a change in feeding vessels, and by the dramatic reduction in incidence of ICC throughout India.

The nature and role of a second factor in the causation of ICC remained unclear, although an inherited defect in copper metabolism was strongly suspected. ICC, however, was not considered to be a straightforward early onset of Wilson disease because ceruloplasmin was consistently normal, and both clinical and histologic recovery was maintained in the long-term despite the withdrawal of D-penicillamine therapy.

##### Indian Childhood Cirrhosis-like Disorder

Descriptions of an ICC-like illness in the West suggested that different mechanisms, including environmental and/or genetic factors, can lead to the same end-stage liver disease: copper-associated childhood cirrhosis [118,119,123]. The conclusion was reached early on that ICC represents a specific form of copper-associated childhood cirrhosis that requires high environmental copper ingestion for its full expression [116].

The initial concept of ICC [116] was challenged by subsequent studies [118,119,120]. While there is agreement that liver diseases of infancy and childhood are rare within the spectrum of these disorders [116,119], only a few subtypes are related to abnormal hepatic copper accumulation [119]. In particular, idiopathic copper toxicosis has now been defined as such a subtype, characterized by distinct clinical and pathologic features; however, its exact etiology is still controversial. Based on a review of the literature and observations of 138 cases endemic to western Austria, the hypothesis was presented that idiopathic copper toxicosis is caused by a synergy of an autosomal-recessive inherited defect in copper metabolism and excess dietary copper [118,119], classified before as an ecogenetic disorder [118]. In line with these considerations is a case report from Germany on a female child of non-consanguineous, healthy German parents who fell ill at the age of 7 months with a progressive liver disease leading to irreversible hepatic failure 3 months later [120]. Histological examination revealed severe liver cell necrosis, excessive Mallory body formation, and veno-occlusive-like changes associated with massive storage of copper, similar to ICC. Chronic copper contamination of drinking water was the only detectable etiological factor. The conclusion was reached that ICC most probably is an environmental disease, also occurring outside the Indian subcontinent, and is likely to be underdiagnosed in the Western world [120].

This severe form of rapidly progressive cirrhosis associated with a marked increase in hepatic copper has been described in children from rural, middle class Hindu families in India [117,119]. Originally termed as ICC, similar clinical cases have been reported worldwide, and this disorder is now referred to as idiopathic childhood cirrhosis or idiopathic copper toxicosis [117,118,119,120,124,125,126,127]. Affected children are diagnosed by two years of age with hepatosplenomegaly, elevation of serum aminotransferase activities, cirrhosis, and elevated liver copper content [117]. The serum ceruloplasmin level in these patients is normal or elevated, suggesting that the defect in biliary copper excretion is distal to the role of ATP7B in this process. Epidemiological investigations of idiopathic childhood cirrhosis further indicated that both genetic and environmental factors may play a role in this disease [25]. Of note, numerous studies have revealed an increase in the copper content of the diet consumed by the affected children, while analysis of some families suggests autosomal recessive inheritance with incomplete penetrance [125]. In support of an underlying defect in hepatic copper excretion, D-penicillamine is effective in many cases, and hepatic transplantation can be curative [117]. Overall, this disease, now called idiopathic childhood cirrhosis syn idiopathic copper toxicosis, is quite different from Wilson disease, traced back alone to genetic variability rather than to additional increased copper consumption through food or beverages.

### 3.5. Symptoms of Wilson Disease

Wilson disease is a multifaceted disorder, difficult to diagnose and often misdiagnosed [89], in part due to physicians’ limited knowledge of its clinical features and a low prevalence, ranging from 1:40,000 to 1:50,000 in the general population [128]. Among 22.5% of patients, the diagnosis was delayed as it was not achieved three years after the initial symptoms were evident, whereby the diagnosis was established at a mean age of 20.4 years with SD 10.6 and a range 4–56 [65]. However, Wilson disease was described even in infants, with an age range from a few months up to 4 years [129].

Apart from the liver, the most implicated organs are the brain, kidneys, eyes, heart, muscles, and bones [89]. Depending on the organ involved and the stage of the disease, patients with Wilson disease may be monosymptomatic, oligosymptomatic, or even polysymptomatic, but some patients may present even as asymptomatic, especially in the initial stages of the disease.

Details of clinical manifestations were perfectly and comprehensively listed [89]: (1) findings related to the liver may include abnormal LTs, chronic active hepatitis, cirrhosis with portal hypertension, and acute liver failure; (2) psychiatric features comprise affect, cognitive, and behavior disorders, as well as depression and psychosis; (3) neurologically, tremor, dysarthria, ataxia, nystagmus, writing problems, and dysphagia with pseudohypersalivation prevail; (4) renal tubular dysfunction; (5) Kayser–Fleischer corneal rings are verified through split-beam examination by an eye specialist; (6) various findings like cardiomyopathy, cardiac arrhythmias, rhabdomyolysis, osteoporosis, osteomalacia, arthritis, and arthralgia. In addition, Coombs-negative hemolytic anemia is a key feature of Wilson disease with undetectable serum haptoglobin, high serum activities of lactate dehydrogenase, and high reticulocyte counts [101,130].

### 3.6. Clinical Presentation of Wilson Disease

Clinical characteristics may include jaundice due to non-immune hemolysis, as evidenced by low blood hemoglobin level of 6.9 g/dL, low serum haptoglobin of <10 mg/dL, high serum activity of lactate dehydrogenase of 2148 U/L, increased serum unconjugated bilirubin of 4.5 mg/dL, and negative Coombs test [98], which typically classifies the non-autoimmune hemolysis of genetic Wilson disease [66], as opposed to autoimmune hemolysis with a positive Coombs test [131]. Copper can also lead to methemoglobinemia, while the diagnosis of acute copper intoxication was confirmed by total serum copper of 874 µg/dL and urine copper of 356 µg/24 h [98]. Notably, in cases of acute exogenous copper sulphate poisoning, intravascular hemolysis commonly starts as early as within the first 24 h since ingestion, and is due to the direct oxidative damage to erythrocyte membranes [99]. The Cu^2+^ ion oxidizes the Fe^2+^ ion in hemoglobin to Fe^3+^, resulting in its conversion to methemoglobin [99,101]. This manifests as cyanosis and the loss of the oxygen carrying capacity of blood [99]. Copper can lead to direct injury of the pro-ximal renal tubules and acute tubular necrosis, with increased serum values of creatinine up to a maximum 621 µmol/L being a sign of renal impairment, attributable also to dehydration, hemoglobinuria, rhabdomyolysis, and sepsis [99].

### 3.7. Diagnostic Approaches

#### 3.7.1. Genetic Screening

In any patient with suspected Wilson disease, first- and second-degree relatives need to be screened for Wilson disease [25,132]. Such family screening facilitates the early diagnosis of Wilson disease, thereby increased the prevalence, as the mean age of patients was significantly lower compared with patients diagnosed at a symptomatic stage of the disease (15.5 vs. 20.4 years; *p* = 0.021) [65].

Mutation analysis of the coding region of ATP7B (except exons 2, 3, and 21) performed in 150 patients with Wilson disease showed no detectable mutations in 15% of patients, and mutations causing the disease were found in 57% of patients on both chromosomes and in 29% of patients on one chromosome [65]. There were no significant differences in the frequency of pathological laboratory test values between the two study cohorts with detectable mutations. Thus, a negative screening results of *ATP7B* gene mutations do not rule out Wilson disease, and the conclusion may be drawn that Wilson disease can develop without an *ATP7B* gene-mutation association. Nevertheless, the search for *ATP7B* gene mutation is helpful for a complete assessment in each patient with suspected Wilson disease, a proposal restricted by current costs and lack of even availability worldwide [89].

There is not a single clinical feature viewed as key diagnostic sign that would help early recognize Wilson disease [65,89]. It should be suspected preferentially in younger patients with jaundice, psychiatric, or neurologic symptoms [89], but it also occurs even at ages above 50 years [65] and in infants, but is uncommon before the age of 3 years [133].

#### 3.7.2. Laboratory Data

Various laboratory tests are used and are under discussion in patients with Wilson disease (Table 3) [25,66,89,129,130,132,133,134,135,136,137,138].

Careful interpretation of laboratory parameters, including the copper-related serum ceruloplasmin, the serum free copper, urinary copper excretion, and hepatic copper content, is recommended, because some are confounded by variables, as discussed in the literature [25,66,89,129,130,132,133,134,135,136,137,138] (Table 3). A caution, also, is warranted regarding routine laboratory parameters of serum LTs that are variable due to confounders and often not helpful to suspecti the diagnosis of Wilson disease (Table 3). As an example, serum activities of AST are often much higher as compared with ALT, likely due to mitochondrial AST released from injured hepatocytes or from lysing erythrocytes containing AST especially in acute liver failure [130,136]. High serum ALT activities up to 800 U/L were commonly found only in younger children with an age range from 4 to 8 years, with low ALT activities in infants of an age < 4 years partially due to hemolytic crisis, and higher ALT values in three other infants [129]. As a result, a realistic R value to determine the liver injury pattern based on LT values is not feasible, as also shown in a recent study providing serum activities for ALT of 46.9 ± 33.8 U/L and for ALP of 158.0 ± 119.4 U/L, but without giving ranges of the normal distribution [139]. In patients with untreated Wilson disease, serum ALP activities may be low, an observation of clinical interest [25,136,137].

Increased plasma levels of inflammatory cytokines and chemokines were found in patients with Wilson disease as compared with healthy controls, an interesting finding but not contributing to a diagnostic biomarker [63]. Serum autoimmune parameters, including anti-nuclear antibodies (ANA), have rarely been reported at the start of the therapy, with higher frequencies during treatment with D-penicillamine [140]. Other laboratory parameters in patients with Wilson disease were abnormal, including serum ceruloplasmin < 0.2 g/L, non-ceruloplasmin-bound serum copper > 25 µg/L, urinary copper excretion > 1.6 µmol/24 h, and liver copper > 250 µg/g dry wt [65].

#### 3.7.3. Diagnostic Algorithm

Mainstream opinion describes Wilson disease a disorder difficult to establish as a firm diagnoses due to variable clinical features and abundant laboratory results that differ in their validity [65,89]. In this context, artificial intelligence (AI) provides a forum whereby complicated processes, including those prevalent in diagnosis of human diseases, are solved by an intelligent diagnostic algorithm using relevant characteristics that give individual scores to be summarized by providing a final score with grades of probability [141]. Using such scoring methods, complex diagnoses of various diseases [140,141,142,143,144,145], like drug induced liver injury (DILI), herb induced liver injury (HILI) [140,141,142,143,144], and autoimmune hepatitis [145], were firmly diagnosed. For instance, DILI and HILI [104,141,142,143,144], were diagnosed by applying the original Roussel Uclaf Causality Assessment Method (RUCAM) of 1993 [142,143] or better performed using the updated RUCAM of 2016 [104]. Wilson disease was diagnosed, for which the Leipzig scoring system of 2003 [146] or the subsequently modified Leipzig soring method of 2019 were successfully applied [147]. Key items of Wilson disease received individual scores, and the sum of these scores classified the suspect Wilson disease as an established diagnosis (score ≥ 4), as possible (score 3), or as very unlikely (score ≤ 2) (Table 4) [147].

Anterior segment optical coherence tomography can be used for the detection and quantification of Kayser–Fleischer rings as the better option compared with a slit-lamp examination by an ophthalmologist [134]. This Modified Leipzig Score was derived from a previous report [147], but for reasons of precision, the score was now revised: under the item of 24-h urinary copper, the previous term “acute hepatitis” was replaced globally by “chronic cholestatic hepato-biliary disease” in Table 4.

The note in the modified Leipzig Scoring System for Wilson disease, as published in 2019 [147] under the item 24-urinary copper (in the absence of acute hepatitis), is not based on any evidence, as plain acute hepatitis is not known among experts for increased urinary copper excretion, as opposed to cholestatic autoimmune hepatitis (AIH) or any other liver disease presenting with cholestatic features like autoimmune cholangitis (AC), primary biliary cholangitis (PBC), or primary sclerosing cholangitis (PSC) [148,149], which may masquerading as Wilson diseases due to high urinary copper excretion [148]. For all four cholestatic liver diseases (AIH, AC, PBC, and PSC), a range of diagnostic parameters are available for exclusion purposes, listed in a worksheet as a checklist of differential diagnosis [104]: AIH type I (ANA, SMA, AAA, SLA/LP, Anti-LSP, Anti-ASGPR, Gamma-globulins), AIH type 2 (Anti-LKM-1 corresponding to CYP 2D6, Anti-LKM-2 corresponding to CYP 2C9, Gamma-globulins), AC (ANA, SMA), PSC (p-ANCA, MRC), and PBC (AMA, Anti PDH-E2). Thus, the term of acute hepatitis was, for reasons of clarity, already now replaced cumulatively by chronic cholestatic liver disease in the modified Leipzig Scoring System (Table 3).

#### 3.7.4. Imaging of the Liver in Patients with Wilson Disease

##### Abdominal Ultrasound

Abdominal ultrasound normally shows an increased liver echogenicity in the initial stages of Wilson disease, primarily in line with a fatty liver disease lacking diagnostic value and missing specificity [150]. Such ultrasound results are difficult to interpret if not correctly differentiated from copper-independent non-alcoholic fatty liver disease (NAFLD), non-alcoholic steatohepatitis (NASH) [151,152], or alcoholic fatty liver disease [153,154,155]. Indeed, clinical conditions are critical since Wilson disease may masquerade as NASH and initiate therapy errors due to incorrect diagnosis [152]. Instead, ultrasound results in Wilson disease are clearly different in already existing cirrhosis [150].

##### Liver Transient Elastography and Shear Wave Elastography

More details were reported using liver transient elastography (TE) or 2-dimensional (2D) shear wave elastography (SWE) [156,157], although the assessment of hepatic manifestations in Wilson disease patients can be improved by combining ultrasound elastography with serologic indices [156]. The use of TE has the advantage of displaying a gradual increase between various stages of hepatic manifestation in Wilson disease, and could significantly discriminate cirrhosis [157].

##### Magnetic Resonance Imaging

Magnetic resonance imaging (MRI) used for the liver in patients with initial stages of Wilson disease is commonly not recommended because copper is not paramagnetic and therefore is not directly detected by MRI, and differs from iron or manganese in this respect. Instead, a brain MRI should be performed in all Wilson disease patients with or without neurological symptoms, because a brain MRI is viewed as the most sensitive imaging tool for the diagnosis of Wilson disease [134] and has been included in the modified Leipzig Scoring System [147]. The procedure is used for the confirmation of cerebral copper accumulation and basal nuclei damage [134]. Neuroimaging patients with neurological Wilson disease signs and symptoms always present with MRI brain alterations. T1-weighted MRI detects atrophic changes, while T2-weighted MRI records signal changes in the putamen [158]. The “face of the giant panda” sign present in the midbrain is considered a characteristic Wilson disease feature [159]. Anterior segment optical coherence tomography can be used for the detection and quantification of Kayser–Fleischer rings as the better option compared to a slit-lamp examination [9,160].

##### Liver Computer Tomography

Hepatic computed tomography (CT) in the early stages of Wilson disease is of little value, and in the late stages, like cirrhosis, it has many CT findings, although most of them are nonspecific [161]. Findings such as hyperdense nodules and honeycomb pattern in a non-contrast-enhanced CT scan and the arterial phase of a triphasic CT scan with lack of hypertrophy of caudate lobes are hallmarks of Wilson disease.

#### 3.7.5. Liver Histology by Light Microscopy

Liver histopathology shows, in the initial stages of Wilson disease, mild unspecific lobular changes, and also occasionally singular apoptotic hepatocytes and spotty necrosis with surrounding lymphocytes, mild macrovesicular steatosis, and ballooned or glycogenated hepatocytes [151]. With the progression of the disease, inflammation increases, leading to fibrosis and ultimately to cirrhosis, with the clinical end-stage being acute liver failure. Increased copper accumulation in the hepatocytes may be visible by histochemical staining with rhodamine or rubeanic acid through direct binding to copper [151,162]. However, absence of a visible copper binding protein does not exclude Wilson disease [151]. At the time of diagnosis, liver histology obtained by liver biopsy in 78 patients showed chronic liver injury in 73% of patients, with fibrosis occurring in 36% and cirrhosis in 37% of patients, while steatosis was found in 54% of the evaluated liver specimens, and a normal liver histology was described in three patients [65].

#### 3.7.6. Liver Ultrastructure

Ultrastructural analysis with electron microscopy of the liver at the early Wilson disease stage of steatosis reveals specific mitochondrial abnormalities [135,163]. Typical findings include variability in size and shape, increased density of the matrix material, and inclusions like lipid and fine granular material that may be copper. The most striking alteration is increased intracristal space with dilatation of the tips of the cristae, creating a cystic appearance [163]. In the absence of cholestasis, these changes are considered to be pathognomonic of Wilson disease. In the later stages of the disease, dense deposits within lysosomes are present. Ultrastructural analysis may be a useful adjunct for diagnosis [135].

## 4. Medical Therapy of Wilson Disease

### 4.1. Dietary Recommendations

Most commonly consumed foods contain trace amounts of copper [66,149], which makes dietary copper restriction less feasible, although food with high copper content should be avoided to support therapy with chelators [66]. Food with a high copper content includes cocoa derived from Owena cocoa (*Theobroma cacao* L.), contaminated with copper from fungicides containing copper, dark chocolate from contaminated cocoa, nuts, raisins, shellfish, oysters, and butchery foodstuffs from the liver and kidneys derived from cattle grazing on grounds with plants possibly contaminated by copper [164,165]. Most important in the clinical dietary context of Wilson disease are products containing considerable amounts of copper like dark chocolate and cocoa, which should not be consumed by patients with Wilson disease [66]. This is in line with the early observation that chocolate contains copper in amounts up to 16.50 ± 1.29 µg/g in a chocolate with 85% cocoa, associated with a linear correlation of the copper content of chocolate to its cocoa content with a correlation coefficient of 0.89, showing that the copper was contributed to the chocolate by the cocoa [165]. Raw cocoa beans from plantations in Nigeria had copper contents ranging from 104 µg/g to 642 µg/g, attributed to the use of copper sulfate as fungicide for disease prevention on these plantations, and was detected in soil and vegetation components [154]. It seems plausible that the high copper levels detected in cocoa (Owena cocoa, *Theobroma cacao* L.) is due to direct contact with the fungicide or via uptake from the soil by horizontal transfer, similar to plant pyrrolizidine alkaloids [166].

### 4.2. Randomized Controlled Trials

There are no prospective studies using randomized controlled trials (RCTs) available with a focus on efficacy and risks of therapeutic approaches in patients with Wilson disease [66], partly attributable to the low disease frequency [167]. However, for any new therapy approach, a control group would be necessary, and this likely withholds from any drug treatment, but this is now too late and would be unethical. Instead, empirical therapy was early applied based on anecdotal reports, personal experience, and considering mechanism leading to the liver injury [66,135]. To achieve reduction of copper excess in the liver, two strategies have to be discussed: one refers to the un commonly proposed prevention through reduction of alimentary copper [9,66,168], and the other one focuses on increasing renal copper elimination as the better alternative [66,134].

### 4.3. Drug Therapy in Wilson Disease

#### 4.3.1. First Line Drug Therapy with D-Penicillamine

Patients with Wilson disease are well treated first line with copper chelators like D-penicillamine that helps remove circulating copper bound to albumin, which facilitates urinary copper excretion via the kidneys [66]. Based on previous considerations that drug treatment with D-penicillamine in patients with Wilson disease may interfere with pyridoxin, co-medication with pyridoxal 5′phosphate, the biologically active form of vitamin B_6_, is commonly prescribed in an amount of 25–50 mg daily [169]. However, a recent preliminary study from France questioned the need of a routine comedication with vitamin B_6_, as blood levels were in the normal range in Wilson disease patients treated with D-penicillamine in the absence of additional vitamin B_6_ supplementation [139]. In this context, it was mentioned that the usual diet contains sufficient amounts of vitamin B_6_, to be supplemented only if food deficient of vitamin B_6_ is consumed. There was also the note that optic neuropathy was observed in four patients in association with a D-penicillamine therapy [139]. However, a temporal association is not necessarily a causal association. Finally, it was argued that little published data on vitamin B6 supplementation for patients treated with D-penicillamine are available, and no consensus recommendation exists [139].

D-penicillamine is initially given at a daily dose of 300 mg and is increased weekly from 300 mg up to 1500 mg/d [66,135,170]. Similar to any other drug treatment for most diseases, the therapy with D-penicillamine is not without ADRs. Initial allergic reactions can be scoped by intermitted drug cessation or concomitant use of corticosteroids [66,139,170]. More serious is an initial deterioration of neurological symptoms occurring in 14% of treated patients, which is hard to reversed and is of unknown etiology [65,66]. Of concern is the development of autoimmunity due to D-penicillamine use, as evidenced by the new appearance of serum anti-nuclear antibodies (ANA) with variable titers, to be differentiated from increased ANA titers occasionally found already before the start of drug therapy attributed to Wilson disease itself [140]. In case of new autoimmunity, switching from D-penicillamine to alternative, second-line therapies using the chelator trientine or zinc is recommended [66].

#### 4.3.2. Rug Therapy with Trientine

For a long time, treatment with trientine has been recommended as a second-line therapy for patients with Wilson disease, although no head-to-head studies for initial treatment were available [66,135,171,172]. Respective previous recommendations are now outdated.

#### 4.3.3. Zinc Therapy

The use of zinc as zinc salts, or the better tolerable zinc–amino-acid-bound medications, is associated with fewer ADRs, except for severe abdominal discomfort that may lead to drug cessation [66]. The recommended dose for adults is 3 × 50 mg daily, and for children or adolescents is 3 × 25 mg daily [173]. Regarding the amelioration of liver injuries, it was deemed inferior to chelators [66,135,170]. Zinc induces intestinal metallothionein, which blocks copper absorption and increases excretion in the stools, resulting in an improvement of symptoms [174]. In addition, two meta-analyses and several large retrospective studies indicate that zinc is equally effective as chelators for the treatment of Wilson disease, with the advantage of an extremely low level of toxicity and only the minor side-effect of gastric disturbance. Thus, zinc may have a role as a first-line therapy in Wilson disease patients with neurological symptoms [135,174], is gaining acceptance for patients with hepatic presentations, and is universally recommended for lifelong patients [174].

#### 4.3.4. Issue of Bis-Choline Tetrathiomolybdate

A short note on Bis-choline tetrathiomolybdate (TTM) is warranted, also known as ALXN1840, a drug in the pipeline of AstraZeneca for treating Wilson disease. It was cut at clinical phase III after results of additional phase II studies were disappointing, not reaching the study aims and following consultations with regulatory agencies, communicated in spring 2023 by Pascal Soriot, AstraZeneca CEO [175]. On theoretical grounds, treatment with TTM appeared promising due to its experimental capacity to increase biliary copper excretion preferentially suitable in Wilson disease patients with neurological symptoms. It was reported that TTM as intracellular chelator removes copper from metallothionine through a process called the stripping effect, whereby this metallothionine remains open for copper binding to remove free copper [66]. However, there were no new data that validated early data from Loong Evans Cinnamon (LEC) rats. Little attention was paid to the fact that copper may be reabsorbed from the intestine via the entero-hepatic circulation, counteracting systemic and hepatic copper depletion. Moreover, any new therapy approach should be more effective compared with previous treatment modalities like D-penicillamine or zinc. Although initial studies with TTM were viewed as positive [66,113,175,176,177], some criticism was communicated regarding low case numbers of the study cohort and quantitative free copper measurement to provide efficient copper removal from the blood as an old drug in a modern design for Wilson disease, raising the question of whether TTM is good for the brain and liver [113]. It will be interesting to see the final information from AstraZeneca as to why clinical phase III was cut [175].

#### 4.3.5. Experimental Therapies

Apart from established clinical therapy using liver transplantation [178,179], key questions focus on new therapeutical approaches that are briefly mentioned with a few examples [115,180,181,182,183,184,185,186,187].

##### Human Hepatocyte Transplantation

Human hepatocyte transplantation may be a new exciting and perfect treatment option in Wilson disease [115,180,181]. It is increasingly used as therapy for patients with hepatic metabolic disorders [180]. In particular, patients with Wilson disease experiencing acute liver failure not responding to chelation therapy may benefit from human hepatocyte transplantation, either as transient support until chelation treatment shows its effect, or as a definitive cure through liver repopulation by healthy donor cells, as shown in animal models of Wilson disease [180]. Although clinical trials of human hepatocyte transplantation have already proven their safety and efficacy in different acute liver failure etiologies, it remains to be similarly demonstrated in patients with acute liver failure due to Wilson disease [180]. Currently, human hepatocyte transplantation is still experimental and does not have established safety and efficacy, as only pilot studies exist. There also is a need for use of treatment to prevent cellular rejection for more than a short-term function of cells. At best, it could replace emergency liver transplantation and lifelong immunosuppression in this cohort.

##### Transplantation of Bone Marrow Cells

Experimental early stage transplantation of bone marrow cells markedly ameliorated copper metabolism and restored liver function in a mouse model of Wilson disease [182,183]. For this purpose, normal bone marrow cells were transplanted in the early staged genetic Wilson disease mouse via caudal vein injection. As a result, transplantation during the early stage significantly corrected copper accumulation, serum AST activities, and serum ceruloplasmin oxidase activity [182]. Comparable results were obtained in an experimental rat model [184].

##### Gene Therapy

Gene replacement therapy theoretically can lead to potential cure of genetic disorders [48,66,185,186,187] through delivering a functional *ATP7B* gene (cDNA) into Wilson disease patients, but major hurdles for its application calls for caution [48]. One consideration is that Wilson disease is a systemic disease with ATP7B expression in multiple cell types. While gene therapy vectors can be delivered systemically and enter multiple cell types, the majority of viral vectors and their resultant expression resides in the liver. Thus, current Wilson disease gene therapy should primarily be considered a liver-specific correction of the disorder. However, while this liver-specific expression is a limitation, case reports of liver transplantation reversing neurologic Wilson disease suggest that liver-specific expression could be sufficient for some Wilson disease patients [48]. In other words, disease could possibly be treated with gene therapy, provided that sufficient transgene expression of ATP7B in hepatocytes are maintained over a longer period. Lentiviral gene transfer that integrates the *ATP7B* gene into the genome has been shown to be effective for ameliorating disease progression in animal models of Wilson disease [185]. However, there are concerns that using such integrative techniques could cause oncogenesis.

Adeno-associated virus is an alternative vector for gene therapy that is gaining popularity due to its ability for direct extra-chromosomal gene transfer to hepatocytes. Using an adeno-associated vector (AAV) encoding human ATP7B complementary DNA (cDNA) in the hepatocytes of the Atp7b-/- Wilson disease mouse, adequate expression of ATP7B was achieved to reduce hepatic copper and prevent liver injury [186]. The enormous size of ATP7B cDNA that affected the vector’s cloning capacity for production of efficient gene transfer remains a substantial problem. To cope with this limitation, a shorter vector coding for a miniATP7B protein was generated that was effective in achieving long-term copper hemostasis in this mouse model of Wilson disease [187]. Based on these studies, AAV gene transfer for the genetic correction of human Wilson disease is planned. Issues to overcome in AAV directed therapy include the potential for using neutralizing antibodies in some patients that have pre-existent antibodies and patients who develop antibodies after administration of AAV. Alternatives to re-introducing the gene via the same AAV vector as hepatocytes turnover also would help maintain the therapy by allowing safe gene transfer into previously treated patients with Wilson disease. There are ongoing clinical trials of AAV-mediated gene transfer of *ATP7B* in humans, in which Wilson disease patients receive a single, peripheral intravenous infusion of UX701 or placebo, as outlined in detail by the US National Institute of Health (NIH) on their website, clinicaltrials.gov.

##### Drug Therapeutics

Methanobactin

Methanobactin is a peptide produced by a bacterium named *Methylosinus trichosporium* with the potential to remove toxic copper from cells [48,115,188,189,190]. The process to experimentally eliminate excess copper out of the liver starts with liver mitochondria injured by accumulated copper, for which methanobactin has a special affinity to bind it [190]. Then, copper bound to methanobactin is ready for transfer to biliary transporters of organic anions, such as the bile salt transporters multidrug resistance (MDR) proteins 1–3, which helps biliary copper excretion, and finally with copper found in the gut of treated animals [190]. Another possible elimination pathway for copper-methanobactin could involve Multidrug and Toxic Compound Extrusion (MATE) transporters that are present at the canalicular membrane in rodent and human liver. Finally, and under major discussion, is a more direct biliary transport that may occur as known from for cell-penetrating peptides like methanobactins, which may be taken up via lysosomal endocytosis and thus help excrete copper from hepatocytes into the biliary areas [190]. Taken together, treatment with methanobactin reversed mitochondrial dysfunction and liver injury in animal models of Wilson disease in the acute phase of copper toxicity [188]. A short course of treatment with methanobactin was shown to effectively remove the excess liver copper content in a Wilson disease rat model, an effect that was sustained for several weeks after stopping treatment [189].

In a second study using this same animal model, animals were fed a high-fat diet that compounded the injury of copper to the liver. Treatment with methanobactin again prevented the mitochondrial injury and improved liver histology. Interestingly, this study highlighted the convergence of pathways for liver injury for steatohepatitis and for Wilson disease with the potential synchronous oxidative injury to the hepatocytes [190]. This has relevance for patients with Wilson disease who may experience exacerbation of their injury if they suffer from concomitant NASH, or have progression of their liver disease despite treatment that is directed against the copper toxicity alone. Despite these promising experimental results on the high copper affinity of methanobactin and its rapid action for copper removal from hepatocytes in animal models which suggests its application for the treatment of patients with Wilson disease and life-threatening courses, RCTs are needed to verify its efficacy and rule out ADRs.

Radical Scavenging Agents

Radical scavenging agents are potential candidates to treat patients with Wilson disease and acute copper liberation, such as in hemolytic crisis, a life-threatening stage triggered by substantial amounts of ROS but yet poorly considered for acute therapeutical intervention. Cimetidine, as well as other H_2_ receptor antagonists like burimamide, ranitidine, famotidine, and tiotidine, are powerful hydroxyl radical scavengers [191,192,193,194] which are able to compete with deoxyribose for the generated hydroxyl radicals [191]. Studies on the part of the cimetidine molecule that might be responsible for its potent hydroxyl radical scavenging activity revealed that the guanidine moiety of cimetidine had little hydroxyl radical scavenging activity, while the other part of the molecule, the methylated imidazole with a sulfur and amino group containing side chain, appeared to be a powerful hydroxyl radical scavenger [192].

In analogy with Wilson disease (Table 1 and Table 2), other types of toxic liver injury are likewise triggered by ROS. Among these are liver injuries caused by ethanol [95,153,154,155,195,196] or carbon tetrachloride, a typical aliphatic halogenated hydrocarbon [105,106,107,197]. With respect to acute intoxications with carbon tetrachloride and other aliphatic hydrocarbons, intravenous cimetidine was successfully applied in patients shortly after acute exposure [105,106,107]. A similar therapeutic approach may be considered in future acute life-threating cases of Wilson disease to fight against powerful molecular chain reactions related to ROS generation.

## 5. Liver Transplantation as Ultima Ratio

Liver transplantation is an option as ultima ratio in untreatable fulminant stages of Wilson disease like acute liver failure or hemolytic crisis. This provides a life-saving approach, comprehensive curative treatment of the disease, restoration of the liver function, and mitigation of portal hypertension [178,179]. Excellent survival rates of one and five years without disease recurrence have now been reported, and in case of deceased donor liver shortage, living relative liver transplantation is an option [178]. Indication for a liver transplantation has to carefully be considered in Wilson disease patients with severe psycho-neurological symptoms [178,179].

## 6. Outcome of Patients with Wilson Disease

### 6.1. Causes of Death

Prognosis is poor in patients with Wilson disease, who missed the benefit of drug therapy, hereby reducing the median life expectancy to 40 years [66,89]. Instead, provided the disease is early diagnosed and patients received an appropriate drug therapy or rare liver transplantation, prognosis is commonly good in Wilson disease [66]. Under these conditions, cumulative survival in 51 patients with Wilson disease was 95% at 33 years after diagnosis and identical with 95% of a control group matched for sex and age, considering that survival was only slightly reduced during the early observation period when liver transplantation was not available for acute liver failure. During the observation period of 33 years and under the D-penicillamine treatment, all clinical signs improved with the exemption of gynecomastia and esophageal varices, associated with the improvement of all neurological symptoms and amelioration of all hematological laboratory test results [66,89].

Before 1948, all patients with Wilson disease died shortly after diagnosis, and since then, despite a wide range of therapeutic options, patients still die [198]. The cause of death analyzed in a study cohort including 67 patients comprising 33 men and34 women out of a series of 300 seen between 1948 and 2000. Patients were classified according to their presentation as neurological with 32 patients, 11 hepatic patients, 10 mixed hepatic/neurological patients, and 6 hemolytic patients. Diagnostic failure was the principal cause of death, but there were multiple other causes, like poor compliance and the development of malignant disease outside the liver after 10 years of follow-up.

### 6.2. Hepato-Biliary Malignancies

#### 6.2.1. Hepatocellular Carcinoma

Hepatocellular carcinoma occurrence was occasionally reported, both in patients with cirrhotic [199,200] and non-cirrhotic disease [201]. It remains unclear whether cirrhosis cases of other etiology, like hepatitis B and C or hemochromatosis, were carefully excluded as variable confounders [199,200,201]. In addition, a temporal relationship between carcinoma occurrence and cirrhosis in Wilson disease cannot be equalized with a definite causal association, but may be the result of a spontaneous event. There also is the note that copper may be protective for hepatocellular carcinoma of patients with Wilson disease [202]. Among 130 patients with Wilson disease during a follow-up of 15 years, two patients developed hepatocellular carcinoma, one despite excellent copper removal after 50 years follow-up, and the other developed it with newly diagnosed Wilson disease [203]. Estimated annual hepatocellular carcinoma risk for all patients was 0.09%, with a subgroup analysis in cirrhotic patients revealing an annual hepatocellular carcinoma risk of 0.14% [20]. The view was expressed that regular surveillance for hepatocellular carcinoma in patients with Wilson disease is not warranted.

#### 6.2.2. Cholangiocarcinoma

The incidence of intrahepatic cholangiocarcinoma for patients with Wilson disease is exceptionally low, even for cirrhotic patients [204,205]. As an example, a 44-year-old male was diagnosed with Wilson disease at the age of 15, and following surgical resection, postoperative histology revealed that the resected portion comprised a regenerative nodule with unexpected hemosiderosis as sign of iron deposit in this copper mediated Wilson disease [205].

## 7. Future Directions

In the past, substantial work has been done on case evaluation standard, clinical diagnosis, and molecular details with focus on the cascade of events leading to the liver injury of Wilson disease. Therefore, the characterization of Wilson disease features has gained a good basic level with fair, balanced perspectives. A few issues remain which require clarification.

### 7.1. Serum Ceruloplasmin

Serum ceruloplasmin is a ferroxidase responsible for 95% copper transport in the blood [206]. It regulates iron metabolism in the copper related Wilson disease [206,207] and is used as a key parameter for diagnostic evaluation of patients with Wilson disease (Table 3 and Table 4) [75,135,147]. However, quality concern exists regarding methodology variability that should be addressed because at least three standard analytical methods currently are in use. Among these are immunoassays, immunoturbidimetry, or immunonephelometry [207]. Serum ceruloplasmin results depend on the method used, and therefore are open for discussion due to their variability. As part of the free-copper calculation, serum ceruloplasmin variability makes a correct determination of the free copper often difficult [66,135].

### 7.2. Serum-Free Copper

Serum-free copper [207,208,209], previously termed more cumbersome as non-ceruloplasmin-bound copper, is complexed to albumin and is only available for excretion if there is significant protein loss by the kidneys [207]. As a perfect parameter assessing the copper status of patients with Wilson disease, serum-free copper is highly esteemed among experts [207,208,209,210], but remained an analytical challenge [209]. Free copper can be quantified by using strong anion exchange chromatography coupled to triple quadrupole inductively coupled plasma mass spectrometry [211] or by measuring the exchangeable copper (CuEXC) that offers a correct view of the free copper overload [212]. It provides information on the spread and severity of Wilson disease. Finally, the parameter of the relative exchangeable copper (REC) (percentage of exchangeable to total serum copper) appreciates the toxic fraction of copper in blood is an excellent biomarker for Wilson disease diagnosis [212]. These two tests, CuEXC and REC, are reliable and non-invasive [212]. To resolve discrepancies observed in the determination of this parameter, quantification of serum-free copper should be standardized and included in corresponding diagnostic algorithms [210]. Additional efforts should be undertaken to validly diagnose fulminant Wilson disease with simple laboratory tests [210].

### 7.3. Standardization of Units

To provide more transparency and comparability, results of copper related parameters should be given in standardized units. Currently, there is a mix of units in terms of mg, g, and µg that is less helpful for comparison purposes (Table 3).

### 7.4. Low Serum ALP Activities

Among patients with Wilson disease, low or undetectable serum ALP activities were occasionally reported and were mostly classified as of unknown causes [25,134,135,136,137,213,214,215,216,217]. This observation deserves further clinical and mechanistic evaluation, considering preliminary observations and proposals. For instance, serum alkaline phosphatase activities are low in those with Wilson-related acute liver failure [25,217], and are associated with hemolytic anemia possibly causing a transient marked hypercupremia with incorporation of Cu^2+^ instead of Zn^2+^ at the ALP active site, yielding an enzyme with reduced activity; however, attempts to simulate this phenomenon in vitro have been unsuccessful [217].

### 7.5. Zinc

Another covered topic in the context of Wilson disease is the mechanistic role of zinc [218,219,220,221,222] as an antagonist of copper because Wilson disease is viewed as a matter primarily of copper but also of zinc [218]. However, studies with low zinc in children with acute liver failure due to Wilson disease have not been duplicated or validated, and the case number of the cohort was not large. Serum zinc levels were significantly lower in patients with Wilson disease experiencing acute liver failure, as compared with those without acute liver failure and correlated with the severity of Wilson disease [219]. Cross-talking of zinc with copper in Wilson disease invites further search and analysis.

### 7.6. Iron

The mechanisms of hepatic transport of iron and copper are intimately related, which also can be observed in Wilson disease [53,56]. The special role of iron in Wilson disease requires special attention in the future, although iron overload has been recognized in patients with Wilson disease without a diagnosis of genetic hemochromatosis in the literature for almost two decades [56]. Iron accumulation in the liver has been reported in patients with Wilson disease [53,54,56,223,224,225,226].

Respective studies provided specific copper and iron patterns, as shown by using ultrastructural identification [53] or liver histology with Russian blue staining [53,224]. Of the chronic liver diseases known to cause iron overload, Wilson disease classically is not listed among them [223]. Results in humans were similarly obtained in animal models mimicking Wilson disease using the novel method of laser ablation inductively coupled plasma mass spectrometry (LA-ICP-MS) [225]. Concurrent measurement of iron in the liver specimen will allow identify early iron overload occurring in patients subjected to standard de-coppering D-penicillamine therapy [226]. This was interpreted as potentially resulting from a blockade of the iron efflux out of the liver to the circulation due to low serum ceruloplasmin in connection with a reduction of ferroxidase activity [53,54].

Whereas LA-ICP-MS cannot be used for evaluating copper and iron presence in the brain [225], other approaches can, like Tesla MRI-histopathological study [227]. Through MRI and transcranial sonography, the presence of both metals in the brain of patients with Wilson disease was confirmed [227,228]. In addition, susceptibility-weighted imaging (SWI) showed an increased signal in the basal ganglia of patients with neurological involvement. Cerebral iron uptake, as measured using ^52^ Fe-citrate positron emission tomography, is also increased. A histopathological study of nine patients confirmed that SWI abnormalities in the caudate, putamen, and pons represent abnormal iron deposition, mainly caused by an increase in iron-containing macrophages, and identified an association between iron accumulation and pathological severity in the putamen [229]. As it stands, iron plays a role not only in the liver, but also in the brain of patients with Wilson disease [227,228,229]. In particular, several converging lines of evidence indicate that the brain iron metabolism is disrupted in Wilson disease. The basis for abnormal iron deposition in Wilson disease is largely unclear; however, copper-dependent enzymes play an important role in iron metabolism. For example, caeruloplasmin, which is also expressed in the brain, exhibits ferroxidase activity, and loss of ceruloplasmin function leads to brain iron accumulation. Since the ceruloplasmin molecule qualifies as a catalyst for redox reactions in plasma, it can oxidize iron from ferrous (2+) to ferric iron (3+), which assists in iron binding to transferrin [227,228]. Interactions of iron and copper in the brain of Wilson disease is of clinical and mechanistic importance and warrants further studies.

Satisfactory progress was seen with focus on ferroptosis and cuproptosis in Wilson disease [10,16,56,57,58,59,60,61,62], but will also require a consideration of the potential involvement of calcicoptosis [230]. Their cross-talking will need additional experimental and clinical refinement.

### 7.7. Acute Hepatitis

Finally, a note is warranted on the modified Leipzig Scoring System for Wilson disease published 2012 [135] or 2019 [147]. Correction is mandatory under the section of 24 h urinary copper regarding the absence of acute hepatitis, and here the term of plain acute hepatitis is misleading, which needs replacement by globally chronic cholestatic hepato-biliary disease, already performed in the currently modified Leipzig Score (Table 4). The mainstream opinion is that experts feel uncomfortable with acute hepatitis because this form of hepatitis is usually not known to increase urinary copper removal.

### 7.8. Copper and Atherosclerosis

Notably, an altered copper bioavailability, as evidenced by copper serum levels, predicts early atherosclerosis as the main cardiovascular risk assessed as the intima-media thickness of carotid artery in obese, non-Wilson disease patients with hepatic steatosis detected by ultrasound [231]. These results led to the assumption that copper may play a contributory role in the development of atherosclerosis connected with fatty liver diseases.

## 8. Conclusions

Wilson disease is a multifaceted disorder that benefits from the modified Leipzig scoring method. This algorithm was created in line with principles of artificial intelligence (AI) to improve diagnostic consistency among difficult processes by selecting the most essential elements that require a sophisticated scoring system. The sum of highly scored key elements provides causality grades of probability. The well-structured diagnostic algorithm of the modified Leipzig scoring system suggests causality gradings from established diagnosis down to possible and very unlikely diagnosis. Its regular use is strongly recommended in future cases of suspected Wilson disease. Combining cases with an established causality grading due to a score of ≥4 helps form homogenous study cohorts available for additional analytical approaches regarding clinical issues and mechanistic challenges. This can expand our views of mechanistic steps leading to the injury as a basis for new therapeutic approaches.

## Figures and Tables

**Figure 1 ijms-25-04753-f001:**
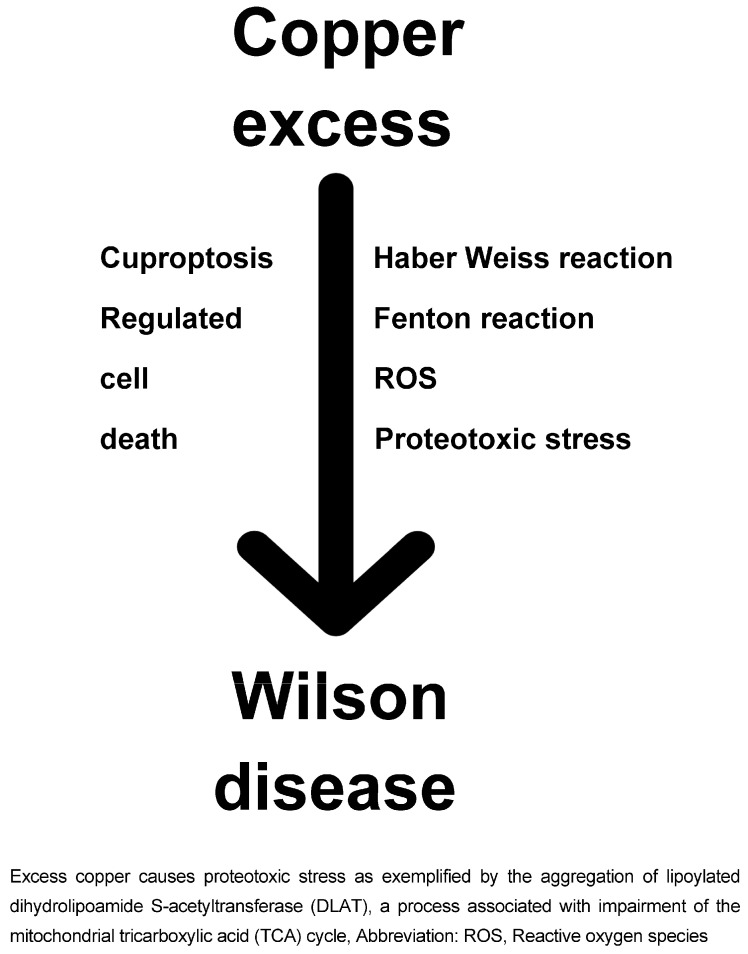
Flow chart of molecular pathway leading from copper excess to Wilson disease.

**Table 1 ijms-25-04753-t001:** Five step sequalae of events leading to the liver injury in patients with Wilson disease.

Cascade of Events	Short Description	References
1. Normal intestinal uptake of copper in healthy individuals and patients with Wilson disease.	In healthy individuals, as well as in patients with Wilson disease, copper absorption from food containing normal amounts of copper proceeds regularly, not under any local control by gene mutation.	Chen et al., 2022 [16], Chaudhry et al., 2023 [25]
2. Normal copper transfer from enterocytes via the portal system to the liver cells in healthy individuals and in patients with Wilson disease.	In patients with Wilson disease, hepatocellular uptake of copper remains unchanged despite ongoing copper accumulation within the liver. There is no regulatory mechanism adjusting the cellular copper uptake according to intracellular copper demand.	Chaudhry et al., 2023 [25], Chen et al., 2020 [26], La Fontaine et al., 2010 [31]
3. In patients with Wilson disease, the excess of copper accumulation within the hepatocytes is due to impaired biliary excretion.	Characteristic of Wilson disease are mutations of *ATP7B* genes that encode the protein ATP7B. This ATP7B transporter has dualistic functions, and both are impaired in patients with Wilson disease: biliary excretion of copper, and synthesis of the copper-containing protein ceruloplasmin.	Chaudhry et al., 2023 [25], Chen et al., 2020 [26], La Fontaine et al., 2010 [31], Lutsenko, 2016 [33], Yu et al., 2018 [40], Weiss et al., 2008 [41], Chang et al., 2017 [42]
4. Copper and its role in generating reactive oxygen species (ROS) as initiator of the liver injury in Wilson disease, possibly in conjunction with iron visible in the liver.	In the liver, intracellular copper in excess produces ROS, similar to iron such as in hemochromatosis. Although most of the ROS production is caused by copper, the liver of patients with Wilson disease normally also contains iron with storage in ferritin is expected. With liver injury, there is acute phase reaction, and this link with inflammation may be the reason for iron present. Most patients do not have hemolytic anemia, which would rather give iron in spleen and hepatic Kupffer cells.	Iakovidis et al., 2011 [10], Tsang et al., 2021 [15], Xue et al., 2023 [52], Shiono et al., 2001 [53], Hayashi et al., 2006 [54], Kadiiska et al., 2002 [55]
5. Role of copper-related cuproptosis and iron-related ferroptosis in the liver injury of patients with Wilson disease. Both reactions depend on free radicals like H_2_O_2_ comprised in ROS.	Major liver injury is caused by intracellular copper binding to lipoylated enzymes in tricarboxylic acid (TCA) cycle that ultimately causes cell death through cuproptosis. Iron accumulated in the liver of patients with Wilson disease likely plays a limited contributary role in the liver injury via ferroptosis. ROS results from the Haber Weiss and Fenton reactions and initiates both, cuproptosis and ferroptosis.	Iakovidis et al., 2022 [10], Chen et al., 2022 [16], Pak et al., 2021 [56], Wang et al., 2022 [57], Tsvetkov et al., 2022 [58], Tang et al., 2022 [59], Gromadzka et al., 2022 [60], Xie et al., 2023 [61], Li et al., 2023 [62]
6. Cross talk among inflammatory cytokines.	In patients with Wilson disease, copper leads to inflammation in the affected organs and tissues. Antibody microarray methods are commonly used to analyze and quantitate plasma levels of inflammatory cytokines. In patients with Wilson disease, their cytokine levels are a response to activation of inflammatory mechanisms, as mostly evidenced by increased expression of interleukins as well as chemokines.	Wu et al., 2019 [63], Kisseleva et al., 2021 [64], Merle et al., 2007 [65], Stremmel et al., 2021 [66], Kelley et al., 1995 [67], Hopkins et al., 1997 [68], Hopkins et al., 1999 [69]
7. Gut microbiome.	The gut microbiome can modify the clinical course of Wilson disease.	Cai et al., 2020 [70]

Abbreviations: ROS, Reactive oxygen species.

**Table 2 ijms-25-04753-t002:** Fourth step of cascade of events triggered by the Fenton and/or Haber Weiss reactions.

Reaction Type	Haber Weiss and Fenton Reactions
Copper-based Haber–Weiss reaction Copper-based Fenton reaction	Cu^2+^ + •O_2_^−^ → Cu^1+^ + O_2_ Cu^1+^ + H_2_O_2_ → Cu^2+^ + OH^−^ + •OH
Iron-based Haber–Weiss reaction Iron-based Fenton reaction	Fe^3+^ + •O_2_^−^ → Fe^2+^ + O_2_ Fe^2+^ + H_2_O_2_ → Fe^3+^ + OH^−^ + •OH
Copper-based Net reaction Iron-based Net reaction	•O_2_^−^ + H_2_O_2_ → OH^−^ + •OH + O_2_ •O_2_^−^ + H_2_O_2_ → OH^−^ + •OH + O_2_

Note: Net reaction represents the result combination of the Haber–Weiss reaction and the Fenton reaction.

**Table 3 ijms-25-04753-t003:** Laboratory test results in patients with Wilson disease.

Laboratory Tests	Normal Range	Test Details in Patients with Wilson Disease	References
Serum ceruloplasmin	0.2–0.5 g/L [134], >0.2 g/L [135]	An amount of <0.1 g/L. Because ceruloplasmin is an acute phase protein it may increase to normal values in co-existing inflammatory diseases and can provide false-negative results [135].	Stremmel et al., 2021 [66], Kasztelan-Szczerbinska et al., 2021 [134], EASL, 2012 [135]
Serum free copper	<150 µg/L [132]	An amount of >250 µg/L [132]. Calculation is required: non-ceruloplasmin-bound free copper (µg/dL) = total serum copper level in µg/dL [132] minus 3 times the level of ceruloplasmin given in mg/L [134], for diagnosis currently not recommended [128]. Serum total copper values of 10–22 μmol/L or 63.7–140.12 μg/dL are acceptable in healthy humans.	Mohr et al., 2019 [132], Kasztelan-Szczerbinska et al., 2021 [134], EASL, 2012 [135]
Serum ALT activity	7–55 U/L, varies from laboratory to laboratory	An amount of >150 U/L in children aged 4–8 years and <50 U/L in patients aged 35 years or older. Normal ALT activities do not rule out Wilson disease. Overall, ALT is of little diagnostic value.	Hayashi et al., 2019 [129], Kasztelan-Szczerbinska et al., 2021 [134]
Serum AST activity	8–48 U/L, varies from laboratory to laboratory	Serum AST activities are often higher than those of ALT, but this higher ratio is of little diagnostic value.	Hassoun et al., 2021 [130], Korman et al., 2008 [136]
Serum ALP activity	40–129 U/L in adults, varies from laboratory to laboratory, with higher ranges in children	Serum ALP activities in adult patients commonly were reported with low levels.	Chaudhry et al., 2023 [25], Korman et al., 2008 [136], Eisenbach 2007 [137]
*ATP7B* gene screening	NA	Sensitivity 90%, specificity 100% [66]. Obligatory test [89]. Although this screening commonly is recommended, restrictions are current costs and the uneven availability worldwide.	Stremmel et al., 2021 [66] Stremmel et al., 2019 [89] Mohr et al., 2019 [132], Kasztelan-Szczerbinska et al., 2021 [134]
Liver copper content	50–249 µg/g dry weight	Correctly >250 µg copper/g dry liver weight [129,132] rather than erroneously >250 mg copper/g dry liver weight [131], which may confirm the diagnosis. However, it is now outdated keeping a strict cutoff of 250 µg copper/g dry tissue. Other problems are confined to invasive procedure, uneven copper distribution, its low tissue level in advanced liver disease causing biopsy sampling errors with false-negative results, and are confounded by high copper levels in in biliary atresia or cholestasis disorders.	Stremmel et al., 2021 [66], Hayashi et al., 2019 [129], Mohr et al., 2019 [132], Kasztelan-Szczerbinska et al., 2021 [134], Roberts et al., 2008 [138]
Urinary copper excretion	0–50 µg/24 h	Usually >100 µg/24 h in adults and 40 µg/24 h in children. This is confirmed to be the most sensitive single screening test for diagnosis. In acute liver failure, false-positive high values are found. Urinary copper excretion typically is above the ranges noted for symptomatic patients, but can be normal early in asymptomatic heterozygotic patients.	Kasztelan-Szczerbinska et al., 2021 [134], EASL, 2012 [135]

Abbreviations: ALP, Alkaline phosphatase; ALT, Alanine aminotransaminase; AST, Aspartate aminotransaminase; NA, not available.

**Table 4 ijms-25-04753-t004:** Modified Leipzig Scoring System for Wilson disease.

Parameter	Score
Kayser Fleischer rings	
Present	2
Absent	0
Serum ceruloplasmin	
>20 mg/dL (normal)	0
0–5 mg/dL	3
6–11 mg/dL	2
11–20 mg/dL	1
24 h urinary copper (in the absence of chronic cholestatic liver disease)	
>100 µg	2
40–100 µg	1
<40 µg	0
Coombs-negative hemolytic anemia with liver disease	
Present	1
Absent	0
Mutational analysis	
On both chromosomes detected	4
On one chromosome detected	1
No mutation detected/test not done	0
Liver biopsy for histology suggestive of Wilson disease with	
Orcein- or rhodamine-positive granules	1
Neurobehavioral symptoms	
Present	2
Absent	1
Typical features on Magnetic Resonance Imaging brain	
Present	1
Absent	0
History of Wilson disease in a family member	
Sibling death from liver disease/neurological disease suggestive of Wilson disease	1
Evaluation	Total score
Diagnosis established	≥4
Diagnosis possible, more tests needed	3
Diagnosis very unlikely	≤2

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
