# Peer review of "Wilson Disease: Copper-Mediated Cuproptosis, Iron-Related Ferroptosis, and Clinical Highlights, with Comprehensive and Critical Analysis Update"

_ijms, 2024, doi:10.3390/ijms25094753_

Round 1
Reviewer 1 Report
Comments and Suggestions for Authors
Dear Authors,
In my opinion your review manuscript covers too many issues related to copper ions in the human body. Thus, the manuscript is not comprehensive nor critical.
First of all, I advise you to divide this manuscript into two (at least) separated manuscripts. Each manuscript should focus only on a few copper-related or Wilson's Disease (WD) aspects.
In this manuscript there is little data on cuproptosis and ferroptosis related to WD. Thus, the title is missilining to the readers. Moreover, 'clinical issues' is not informative.
The introduction section presents few irrelevant data on metal copper characteristics, while the real 'introduction' section is missing. On the other side, in chapters 2 and 3 we can find a huge set of data, which are mostly irrelevant, but could be used to combine a nice 'introduction' section. Moreover, the long and detailed description of copper biochemistry (chapters 2 and 3) could be replaced by the simple Schemes and Figures.
Tables are used in the wrong way. The use of Tables should be re-evaluated.
The entire review manuscript is messy, mixing the epidemiological studies with experimental in vitro data and the use of AI for the diagnosis.
For all these reasons, I do not recommend the manuscript to the publication process.
Best regards
Reviewer
Comments on the Quality of English LanguageThis manuscript needs English grammar and stylistics revisions
Author Response
Point-by-point reply to comments of Reviewer 1
Dear Reviewer,
Thank you for your comments to improve the quality of our paper. We regret that the submitted version did not fulfill your expectations.
Here is our point-by point response:
- We agree that many issues are covered. However, this is needed for a careful analysis as mentioned in the title. Reviewer 2 even asked for many more details to be incorporated.
- We prefer keeping all data in a single paper rather than having two separate reports, thereby avoiding overlapping.
- Your point on cuproptosis and ferroptosis is well taken, we have expanded this section 3.2.5.1. and 3.2.5.2., shown in color. Clinical issues now are replaced by clinical highlights.
- By purpose, the introduction was kept short to avoid redundancies with our outlines in chapter 2 and 3.
- The detailed description of copper biochemistry remained at the current place and cannot be placed in the introduction nor in Tables or Schemes.
- We checked the Tables and found them in a good way.
- Mixing epidemiological data with experimental ones will help readers understand our proposals. AI was included to follow current worldwide views on other relevant topics.
In essence, we incorporated your suggestions as far as they substantially improved our paper. You may check the revised version that clarified many uncertainties. We hope you now can appreciate our work on a hot topic related to Wilson disease and live with the newly edited text to be endorsed for publication.
Kind regards,
Rolf Teschke and Axel Eickhoff
Reviewer 2 Report
Comments and Suggestions for Authors
The introduction of the concepts of cuproptosis and ferroptosis and their potential contribution to injury to the liver in Wilson disease is welcome. The current manuscript is interesting but is all over the place and unfocused with respect to how to integrate these important concepts - and indeed this gets lost in the very extensive review on Wilson disease. A reorganizational rewrite will help in this direction.
Separately, the following comments and suggestions are made to the authors:
Abstract – “inheritable malfunction or missing ATP7B genes” - would refer to the ATP7B protein here, the gene itself is not the malfunctioning agent other than to alter the protein.
Abstract – “extruded from liver cells due to limited storage capacity” – copper is released due to transport or from injured cells, please find a better way to express this. Extruded is not the right word to utilize here.
Abstract – “up to cirrhosis” can be removed from this sentence as you don’t need to be specific about the varied liver injury, just stating in this sentence organs that are pathologically affected by excess copper.
Abstract – liver transplant is not used to treat hemolytic crisis unless it is accompanying acute liver failure due to WD. Can leave the untreatable hemolytic crisis out of this sentence and then it would read to use liver transplant for end stage liver disease or acute liver failure.
P2 Introduction – great first paragraph about origin of heavy metals, but poor transition to why Wilson disease was looked at. At least preface that there are disorders of metabolism that arise from mishandling of heavy metals.
P2 Copper deficiency and anemia – may be explicit and mention sideroblastic anemia
P2 Be careful in discussion of Menkes disease, it is not always a pure copper deficiency but rather a maldistribution of copper in some tissues, as noted for enterocytes. Similarly, it is not animals with copper deficiency that have fatal course, but those with defects in transport of copper to critical sites for development and function.
P3 Discussion of copper in GI tract – would keep focus on ATP7A here, movement of ATP7B does occur with low and high copper states, but is impactful in the liver where it moves copper to the apical membrane. Please use apical and basolateral terms in describing copper movement in the enterocytes as well.
P3 It is not right to refer to the binding properties of albumin and other molecules in the circulation as “chaperones” as these are molecules that bind and release copper based on binding affinity of sites of the molecules, not with specificity of intracellular target (ie like ATOX or COX discussed next section). This writer is skeptical of “transcuprein” as well – the cDNA found by Linder encoded a macroglobulin. No further identification in humans as “transcuprein” follows this work in 2007. Would suggest to just say macroglobulins.
P3 Please be much more careful in the description of compartments where the transporters function. ATP7B helps ceruloplasmin acquire copper in the trans golgi, it does not pump copper into the bloodstream. ATP7B does help deliver copper to the apical bile canaliculus for export of copper via the biliary tract. Similarly, discussion of other molecules transporting copper into different organelles/compartments for use or export should be considered by the authors.
P3,4 Please integrate discussion in 2.6 into the prior section, or move that section earlier in the manuscript.
P4 and throughout – convention italicizes the gene, leaves the protein normal. Ie ATP7B gene vs ATP7B protein. Please use this convention to make it clear which is being referred to.
P4 would avoid saying apoceruloplasmin is rapidly degraded – without a strict numeric difference, easier to just say its circulating half life is shorter than the holoprotein with proper complement of copper (leading to lower concentration of the protein in the circulation).
P5. Table 1 B would not call ATP7B an enzyme.
Table 1 4. Iron in hepatocytes is also normal, storage in ferritin is expected. With liver injury there is acute phase activation, and this link with inflammation and iron may be the reason for iron present. Most patients do not have hemolytic anemia, which would rather give RE iron in spleen and Kupffer cells.
Table 1 6. Cytokines are not dysregulated, but rather their levels are a response to activation of inflammatory mechanisms.
P6. Copper is not delivered by ceruloplasmin to hepatocytes. Transcuprein may not even exist as mentioned. Albumin in the portal circulation is the main delivery mechanism for copper absorbed by enterocytes to reach hepatocytes.
P6 3.2.3 would convey more clearly that liver is the initial site for accumulation based on the ATP7B functional impairment in that tissue, and other organ accumulation afterwards.
P6 3.2.4 Iron accumulation is not generally a major feature of WD just due to a low ceruloplasmin level – most patients do have some circulating level of oxidase active ceruloplasmin. Though there was a review put on in this direction, there is few data to back this up.
Separately would remove the words “and to be on the safe side”
P7 3.2.51 and 3.2.52
Ferroptosis and cuproptosis are forms of regulated cell death, and the reader would benefit from an introduction to the definition of this overarching term and the relationship of both to the conceptual framework about whether necroptosis or aptotosis and separately autophagy (that is not well mentioned here), are important for WD. Reference 63 that was included for the manuscript is a very detailed discussion of some of this, and I am glad this was included. Perhaps a figure depicting the overlapping processes in the liver, with inclusion of cytokine activation as well, might help readers.
P8 re cytokines. The presence of levels different than controls don’t indicate “dysregulation”, and I would caution authors that this term implies a different response than expected for standard stimuli to cytokine production. Please just note the differences, and reserve this term only if the data showed deviation in expected responses.
P9 3.3 Again would remove “extruded” as a term to denote copper export from liver after storage capacity is exceeded.
Would mention specifically that the natural clinical course could include neurologic and or psychiatric disease on the basis of what the authors describe as liver unrelated symptoms and clinical features.
P9 3.4 When stating differential diagnosis here, you are limiting this to a differential to other states of copper overload, not necessarily to just Wilson disease. That can be clarified better. If that is the path the authors wished to take, other considerations include glycosylation defects and PFIC variants with cholestasis. Otherwise in general the differential is other liver disease and other neuro/psych presentations of disease.
3.4.1 re Acute copper poisoning
“About 60% of the ingested dose is absorbed in the gastrointestinal tract and attached mostly to ceruloplasmin (95%), whereas free copper binds to albumin forming its toxic form (104).” The toxic ingested dose is unlikely to be attached to ceruloplasmin – what is absorbed will bind to albumin and much will go through the liver and into bile, and excess out in the urine where copper is usually only minimally excreted, but in this setting urine copper can be significantly elevated. A normal individual is not lacking copper for incorporation into ceruloplasmin.
If the authors only wished to introduce this as a possible differential diagnosis, then there is no need for a prolonged discussion of thoughts about treatment. It is interesting, but irrelevant unless they wished to make points about commonality of liver injury mechanisms, if that is known.
P13 3.71
I think the authors meant to say “ a negative screening for ATP7B mutations does not rule out”…..
While it is nice to say searching for mutations in ATP7B is obligatory, the authors should consider current costs and availability which are not even worldwide and temper the statement a bit. Same holds for notation on table on P 14.
P14. Liver copper is not “outdated” – keeping a strict cutoff of 250 mcg/g dry wt tissue is outdated.
P14 Urinary copper excretion is typically above the ranges noted in “symptomatic” patients but can be normal early on in asymptomatic patients.
P15 sentences with “cautionary” and “Back to real world” are incomplete and need rephrasing.
P18 would replace “visible copper” with “visible copper binding protein”
P18 4.3.2 trientine as second line therapy is not true any longer for maintenance therapy (no studies head to head for initial treatment are available) given findings in the Chelate study (Lancet Gastroenterol Hepatol. 2022 Dec;7(12):1092-1102. )
P20 4.3.4 re TTM unfortunately there were no new references that validated early data from LEC rats that TTM caused increased biliary excretion. The concept of the tripartite complex detoxifying elevated bioavailable copper was the rationale for the use of the drug, especially in neurologic patients. The discussion on this medication should include this concept for indeed this was the main reason for considering the TTM trial.
P20 4.3.5.1 human hepatocyte transplantation is still experimental and does not have “proven” safety and efficacy. Only pilot studies exist. Discussion of the need for use of treatment to prevent cellular rejection for more than short term function of cells should be included. Concept of turning autologous stem cells into hepatocytes for transplant to avoid this may also be mentioned.
P21 4.4.5.3 note that there are ongoing clinical trials of AAV mediated gene transfer of ATP7B in humans. See clinicaltrials.gov
4.3.5.4.1 methanobactin – suggest mentioning the study showing mechanism of action to increase biliary copper excretion and reduce hepatic copper. Partly included in ref 196 and in the newer Gastroenterology paper.
P23 serum free copper - mention of new techniques, CuEx, NCC determination is recommended. (Anal Chim Acta. 2020 Feb 15;1098:27-36., Ann Transl Med. 2019 Apr;7(Suppl 2):S70.)
P24 7.5 zinc avoid value judgements – strike “mediocre” . Study of low zinc in acute liver failure in a pediatric cohort has not been duplicated or validated. The study did not include a large number of patients.
P24 Copper is not paramagnetic and is not directly detected by MRI and differs from iron or manganese in this regard.
Comments on the Quality of English Languageneeds further editing, some of the recommendations are included in comments to authors, but further editing is also in order.
Author Response
Dear Reviewer 2,
Your constructive superb expert recommendations and clear advice to improve the quality of our paper are highly appreciated.
Kind regards,
Rolf Teschke and Axel Eickhoff
Here is our point-by point reply on your comments and suggestions for authors:
The introduction of the concepts of cuproptosis and ferroptosis and their potential contribution to injury to the liver in Wilson disease is welcome. The current manuscript is interesting but is all over the place and unfocused with respect to how to integrate these important concepts - and indeed this gets lost in the very extensive review on Wilson disease. A reorganizational rewrite will help in this direction. REWRITE WAS DONE.
Abstract – “inheritable malfunction or missing ATP7B genes” - would refer to the ATP7B protein here, the gene itself is not the malfunctioning agent other than to alter the protein. DONE
Abstract – “extruded from liver cells due to limited storage capacity” – copper is released due to transport or from injured cells, please find a better way to express this. Extruded is not the right word to utilize here. DONE
Abstract – “up to cirrhosis” can be removed from this sentence as you don’t need to be specific about the varied liver injury, just stating in this sentence organs that are pathologically affected by excess copper. DONE
Abstract – liver transplant is not used to treat hemolytic crisis unless it is accompanying acute liver failure due to WD. Can leave the untreatable hemolytic crisis out of this sentence and then it would read to use liver transplant for end stage liver disease or acute liver failure. DONE
P2 Introduction – great first paragraph about origin of heavy metals, but poor transition to why Wilson disease was looked at. At least preface that there are disorders of metabolism that arise from mishandling of heavy metals. DONE
P2 Copper deficiency and anemia – may be explicit and mention sideroblastic anemia. DONE
P2 Be careful in discussion of Menkes disease, it is not always a pure copper deficiency but rather a maldistribution of copper in some tissues, as noted for enterocytes. Similarly, it is not animals with copper deficiency that have fatal course, but those with defects in transport of copper to critical sites for development and function. DONE
P3 Discussion of copper in GI tract – would keep focus on ATP7A here, movement of ATP7B does occur with low and high copper states, but is impactful in the liver where it moves copper to the apical membrane. Please use apical and basolateral terms in describing copper movement in the enterocytes as well. DONE
P3 It is not right to refer to the binding properties of albumin and other molecules in the circulation as “chaperones” as these are molecules that bind and release copper based on binding affinity of sites of the molecules, not with specificity of intracellular target (ie like ATOX or COX discussed next section). This writer is skeptical of “transcuprein” as well – the cDNA found by Linder encoded a macroglobulin. No further identification in humans as “transcuprein” follows this work in 2007. Would suggest to just say macroglobulins. DONE
P3 Please be much more careful in the description of compartments where the transporters function. ATP7B helps ceruloplasmin acquire copper in the trans Golgi, it does not pump copper into the bloodstream ATP7B does help deliver copper to the apical bile canaliculus for export of copper via the biliary tract . DONE. Similarly, discussion of other molecules transporting copper into different organelles/compartments for use or export should be considered by the authors. DONE already previously on top under 2.5, not redone now to avoid redundancy but backed up by ref 33-36 at least.
P3,4 Please integrate discussion in 2.6 into the prior section, or move that section earlier in the manuscript. DONE
P4 and throughout – convention italicizes the gene, leaves the protein normal. Ie ATP7B gene vs ATP7B protein. Please use this convention to make it clear which is being referred to. DONE
P4 would avoid saying apoceruloplasmin is rapidly degraded – without a strict numeric difference, easier to just say its circulating half-life is shorter than the holoprotein with proper complement of copper (leading to lower concentration of the protein in the circulation). DONE at 3.1
P5. Table 1 B would not call ATP7B an enzyme. REMOVED: enzyme
Table 1 4. Iron in hepatocytes is also normal, storage in ferritin is expected. With liver injury there is acute phase activation, and this link with inflammation and iron may be the reason for iron present. Most patients do not have hemolytic anemia, which would rather give RE iron in spleen and Kupffer cells. DONE
Table 1 6. Cytokines are not dysregulated, but rather their levels are a response to activation of inflammatory mechanisms. DONE
P6. Copper is not delivered by ceruloplasmin to hepatocytes. Transcuprein may not even exist as mentioned. Albumin in the portal circulation is the main delivery mechanism for copper absorbed by enterocytes to reach hepatocytes. YES, CORRECT
P6 3.2.3 would convey more clearly that liver is the initial site for accumulation based on the ATP7B functional impairment in that tissue, and other organ accumulation afterwards. DONE
P6 3.2.4 Iron accumulation is not generally a major feature of WD just due to a low ceruloplasmin level – most patients do have some circulating level of oxidase active ceruloplasmin. Though there was a review put on in this direction, there is few data to back this up. DONE
Separately would remove the words “and to be on the safe side” DONE
P7 3.2.51 and 3.2.52 Ferroptosis and cuproptosis are forms of regulated cell death, and the reader would benefit from an introduction to the definition of this overarching term and the relationship of both to the conceptual framework about whether necroptosis or apoptosis and separately autophagy (that is not well mentioned here), are important for WD. Reference 63 that was included for the manuscript is a very detailed discussion of some of this, and I am glad this was included. DONE. Perhaps a figure depicting the overlapping processes in the liver NOT DONE as too complex, with inclusion of cytokine activation as well, might help readers DONE already under 3.2.6.
P8 re cytokines. The presence of levels different than controls don’t indicate “dysregulation”, and I would caution authors that this term implies a different response than expected for standard stimuli to cytokine production. Please just note the differences, and reserve this term only if the data showed deviation in expected responses. DONE
P9 3.3 Again would remove “extruded” as a term to denote copper export from liver after storage capacity is exceeded. DONE
Would mention specifically that the natural clinical course could include neurologic and or psychiatric disease on the basis of what the authors describe as liver unrelated symptoms and clinical features. DONE
P9 3.4 When stating differential diagnosis here, you are limiting this to a differential to other states of copper overload, not necessarily to just Wilson disease. That can be clarified better. If that is the path the authors wished to take, other considerations include glycosylation defects and PFIC variants with cholestasis. Otherwise in general the differential is other liver disease and other neuro/psych presentations of disease. DONE
3.4.1 re Acute copper poisoning “About 60% of the ingested dose is absorbed in the gastrointestinal tract and attached mostly to ceruloplasmin (95%), whereas free copper binds to albumin forming its toxic form (104).” The toxic ingested dose is unlikely to be attached to ceruloplasmin – what is absorbed will bind to albumin and much will go through the liver and into bile, and excess out in the urine where copper is usually only minimally excreted, but in this setting urine copper can be significantly elevated. A normal individual is not lacking copper for incorporation into ceruloplasmin. DONE
If the authors only wished to introduce this as a possible differential diagnosis NO , then there is no need for a prolonged discussion of thoughts about treatment AUTHORS PREFER KEEPING THAT. It is interesting, but irrelevant unless they wished to make points about commonality of liver injury mechanisms, if that is known.
P13 3.71 I think the authors meant to say “ a negative screening for ATP7B mutations does not rule out”…DONE
While it is nice to say searching for mutations in ATP7B is obligatory, the authors should consider current costs and availability which are not even worldwide and temper the statement a bit. Same holds for notation on table on P 14. DONE
P14. Liver copper is not “outdated” – keeping a strict cutoff of 250 mcg/g dry wt tissue is outdated. DONE in Table 3.
P14 Urinary copper excretion is typically above the ranges noted in “symptomatic” patients but can be normal early on in asymptomatic patients. DONE Table 1
P15 sentences with “cautionary” and “Back to real world” are incomplete and need rephrasing. DONE
P18 would replace “visible copper” with “visible copper binding protein” DONE
P18 4.3.2 trientine as second line therapy is not true any longer for maintenance therapy (no studies head to head for initial treatment are available) given findings in the Chelate study (Lancet Gastroenterol Hepatol. 2022 Dec;7(12):1092-1102. ) DONE
P20 4.3.4 re TTM unfortunately there were no new references that validated early data from LEC rats that TTM caused increased biliary excretion. The concept of the tripartite complex detoxifying elevated bioavailable copper was the rationale for the use of the drug, especially in neurologic patients. The discussion on this medication should include this concept for indeed this was the main reason for considering the TTM trial. DONE
P20 4.3.5.1 human hepatocyte transplantation is still experimental and does not have “proven” safety and efficacy. Only pilot studies exist. Discussion of the need for use of treatment to prevent cellular rejection for more than short term function of cells should be included. Concept of turning autologous stem cells into hepatocytes for transplant to avoid this may also be mentioned. DONE
P21 4.4.5.3 note that there are ongoing clinical trials of AAV mediated gene transfer of ATP7B in humans. See clinicaltrials.gov DONE at end of 4.3.5.3.
4.3.5.4.1 methanobactin – suggest mentioning the study showing mechanism of action to increase biliary copper excretion and reduce hepatic copper. Partly included in ref 196 and in the newer Gastroenterology paper. DONE
P23 serum free copper - mention of new techniques, CuEx, NCC determination is recommended. (Anal Chim Acta. 2020 Feb 15;1098:27-36.,) DONE
P24 7.5 zinc avoid value judgements – strike “mediocre” . Study of low zinc in acute liver failure in a pediatric cohort has not been duplicated or validated. The study did not include a large number of patients. DONE
P24 Copper is not paramagnetic and is not directly detected by MRI and differs from iron or manganese in this regard. DONE under 3.7.4.3.
Comments on the Quality of English Language needs further editing, some of the recommendations are included in comments to authors, but further editing is also in order. DONE with the help of an US native with good English.
Reviewer 3 Report
Comments and Suggestions for Authors
1. In the abstract, the author stated that "copper originally formed in the universe...". All the elements formed in the universe, copper is not an exception. Instead of this statement author should rephrase the statement.
2. In the introduction the author didn't discuss the Wilson disease, which is the theme of this review article.
3. In 2.1 section, the author should mention the tissue name, in which tissue, copper found in which form.
4. In section 2.2, 1st paragraph, statements are not clear, author should rephrase the paragraph.
5. In section 2.2 second paragraph, the author mentioned two genes ATP7A and ATP7B. The author should furnish details of the gene how it works and how linked.
6. In the section Physiological Role of Copper: The author is unable to justify the paragraph with the section heading. Lacking the flow of concepts in the paragraph.
7. In the Second section the author discusses the copper in the human body, their role, and metabolism. However, the review article title contains copper as well as iron both in the context of Wilson disease. The author should amend the iron in this section.
8. The author should rephrase the paragraphs to make a coherent storyline centered around the Wilson disease.
9. This review focused on Wilson's disease, however, it appears in the text in a later section. The author should introduce the basics of the Wilson disease section before the role of copper and then their causative agents.
10. In section 3.1, the author didn't discuss which type of mutations occurred in the Wilson disease. The author should provide the type of mutations and their locations in the genes as per the study reports.
11. There is inconsistency in the notations of gene ATP7B/ATPB7/ATB7/ATP7b. The author should justify the uses of different notations of the same gene or all are different genes.
12. The author should mention the acceptable value of copper in a healthy human being instead of only excessive accumulations.
13. Symptoms of Wilson disease should be placed with the basics of Wilson disease.
Author Response
Dear Reviewer 3,
Thank you for your constructive recommendations.to improve the quality of our paper.
Kind regards,
Rolf Teschke and Axel Eickhoff
Here is our point-by point reply on your comments and suggestions for authors:
- In the abstract, the author stated that "copper originally formed in the universe...". All the elements formed in the universe, copper is not an exception. Instead of this statement author should rephrase the statement. DONE by adding: all.
- In the introduction the author didn't discuss the Wilson disease, which is the theme of this review article. DONE
- In 2.1 section, the author should mention the tissue name, in which tissue, copper found in which form.DONE under 3.1.
- In section 2.2, 1st paragraph, statements are not clear, author should rephrase the paragraph.DONE by adding: and plays a role in.
- In section 2.2 second paragraph, the author mentioned two genes ATP7A and ATP7B. The author should furnish details of the gene how it works and how linked. DONE under 2.5 third para
- In the section Physiological Role of Copper: The author is unable to justify the paragraph with the section heading. Lacking the flow of concepts in the paragraph. REPHRASED
- In the Second section the author discusses the copper in the human body, their role, and metabolism. However, the review article title contains copper as well as iron both in the context of Wilson disease. The author should amend the iron in this section. DONE by including: also with iron.
- The author should rephrase the paragraphs to make a coherent storyline centered around the Wilson disease. DONE
- This review focused on Wilson's disease, however, it appears in the text in a later section. The author should introduce the basics of the Wilson disease section before the role of copper and then their causative agents. NOT POSSIBLE, as according to IJMS rules, emphasis must be given to molecular and mechanistic events, topics therefore covered in the first part.
- In section 3.1, the author didn't discuss which type of mutations occurred in the Wilson disease. The author should provide the type of mutations and their locations in the genes as per the study reports. DONE
- There is inconsistency in the notations of gene ATP7B/ATPB7/ATB7/ATP7b. The author should justify the uses of different notations of the same gene or all are different genes. DONE
- The author should mention the acceptable value of copper in a healthy human being instead of only excessive accumulations.DONE in table 3 third row
- Symptoms of Wilson disease should be placed with the basics of Wilson disease. NOT REQUIRED, as section of 3.5. Symptoms of Wilson disease is already under the heading of 3. Basics of Wilson disease.
Round 2
Reviewer 1 Report
Comments and Suggestions for Authors
Dear Authors,
I do not agree with your revisions, which are not in line with my suggestions.
Best regards
Reviewer
Comments on the Quality of English LanguageExtensive english editing is needed
Author Response
Reviewer 1
Comments and Suggestions for Authors
Dear Authors,
I do not agree with your revisions, which are not in line with my suggestions.
Dear Reviewer,
Thank you for attempts to improve our paper.
Here is our answer:
SORRY, your suggestions to spit our paper into 2 papers is not really a good idea and not acceptable for us. Splitting was not recommended by the two other reviewers. So, we would like to follow the majority vote.
Kind regards,
Rolf Teschke and Axel Eickhoff
Reviewer 3 Report
Comments and Suggestions for Authors
1. In the abstract, the statement "originally formed in the universe" does not make any sense, copper is just an example. The author should rephrase the statement.
2. As Wilson disease is the central theme of this review article, the author should give a brief introduction about Wilson disease, and then how ATP7B gene mutation is linked with it, and finally how copper and iron are associated with Wilson.
3. The author should add a flow chart or cartoon to show Wilson disease-associated molecular pathway.
Author Response
Reviewer 3
Thank you for comments and suggestions for authors. Our answers are included below.
- In the abstract, the statement "originally formed in the universe" does not make any sense, copper is just an example. The author should rephrase the statement. DELETED
- As Wilson disease is the central theme of this review article, the author should give a brief introduction about Wilson disease, and then how ATP7B gene mutation is linked with it, and finally how copper and iron are associated with Wilson. DONE in introduction, shown in yellow.
- The author should add a flow chart or cartoon to show Wilson disease-associated molecular pathway. DONE in a new para 3.2.8.
Kind regards,
Rolf Teschke and Axel Eickhoff
